# Spatial modulation of individual behaviors enables an ordered structure of diverse phenotypes during bacterial group migration

**Yang Bai**[1†], **Caiyun He**[1,2†], **Pan Chu**[1,2], **Junjiajia Long**[3], **Xuefei Li**[1,2], **Xiongfei Fu**[1,2*]

[1]CAS Key Laboratory for Quantitative Engineering Biology, Guangdong Provincial Key Laboratory of Synthetic Genomics, Shenzhen Institute of Synthetic Biology, Shenzhen Institutes of Advanced Technology, Chinese Academy of Sciences, Shenzhen, China; [2]University of Chinese Academy of Sciences, Beijing, China; [3]Yale University, Department of Physics, New Haven, United States

**Abstract** Coordination of diverse individuals often requires sophisticated communications and high-order computational abilities. Microbial populations can exhibit diverse individualistic behaviors, and yet can engage in collective migratory patterns with a spatially sorted arrangement of phenotypes. However, it is unclear how such spatially sorted patterns emerge from diverse individuals without complex computational abilities. Here, by investigating the single-cell trajectories during group migration, we discovered that, despite the constant migrating speed of a group, the drift velocities of individual bacteria decrease from the back to the front. With a Langevin-type modeling framework, we showed that this decreasing profile of drift velocities implies the spatial modulation of individual run-and-tumble random motions, and enables the bacterial population to migrate as a pushed wave front. Theoretical analysis and stochastic simulations further predicted that the pushed wave front can help a diverse population to stay in a tight group, while diverse individuals perform the same type of mean reverting processes around centers orderly aligned by their chemotactic abilities. This mechanism about the emergence of orderly collective migration from diverse individuals is experimentally demonstrated by titration of bacterial chemoreceptor abundance. These results reveal a simple computational principle for emergent ordered behaviors from heterogeneous individuals.

**\*For correspondence:**
xiongfei.fu@siat.ac.cn

[†]These authors contributed equally to this work

**Competing interest:** The authors declare that no competing interests exist.

## Introduction

Collective group migration, as an important class of coordinated behaviors, is ubiquitous in living systems, such as navigation, foraging, and range expansion (*Krause et al., 2002*; *Partridge, 1982*; *Sumpter, 2010*). In the presence of individual heterogeneity, the migrating group often exhibits spatially ordered arrangements of phenotypes (*Krause et al., 2002*; *Parrish and Edelstein-Keshet, 1999*; *Partridge, 1982*; *Sumpter, 2010*). In animal group migration, individual behavioral abilities (e.g., directional sensitivity) would result in social hierarchy, which further drives the spatial arrangement in a coordinated group (*Couzin et al., 2005*). At the same time, spatial arrangements can lead to different costs and benefits for the individuals participating in the group migration (*Krause, 1994*; *Parrish and Edelstein-Keshet, 1999*; *Partridge, 1982*). Participating individuals must follow disciplinary rules to organize themselves into coordinated patterns while on the move, which requires complex computational abilities to interact with the group and the environment (*Couzin and Krause, 2003*; *Couzin et al., 2002*; *Vicsek and Zafeiris, 2012*). Therefore, understanding how individuals of

**eLife digest** Organisms living in large groups often have to move together in order to navigate, forage for food, and increase their roaming range. Such groups are often made up of distinct individuals that must integrate their different behaviors in order to migrate in the same direction at a similar pace. For instance, for the bacteria *Escherichia coli* to travel as a condensed group, they must coordinate their response to a set of chemical signals called chemoattractants that tell them where to go.

The chemoattractants surrounding the bacteria are unequally distributed so that there is more of them at the front than the back of the group. During migration, each bacterium moves towards this concentration gradient in a distinct way, spontaneously rotating its direction in a 'run-and-tumble' motion that guides it towards areas where there are high levels of these chemical signals. In addition to this variability, how well individual bacteria are able to swim up the gradient also differs within the population. Bacteria that are better at sensing the chemoattractant gradient are placed at the front of the group, while those that are worst are shifted towards the back. This spatial arrangement is thought to help the bacteria migrate together. But how *E. coli* organize themselves in to this pattern is unclear, especially as they cannot communicate directly with one another and display such diverse, randomized behaviors.

To help answer this question, Bai, He et al. discovered a general principle that describes how single bacterial cells move within a group. The results showed that *E. coli* alter their run-and-tumble motion depending on where they reside within the population: individuals at the rear drift faster so they can catch up with the group, while those leading the group drift slower to draw themselves back. This 'reversion behavior' allows the migrating bacteria to travel at a constant speed around a mean position relative to the group.

A cell's drifting speed is determined by how well it moves towards the chemoattractant and its response to the concentration gradient. As a result, the mean position around which the bacterium accelerates or deaccelerates will vary depending on how sensitive it is to the chemoattractant gradient. The *E. coli* therefore spatially arrange themselves so that the more sensitive bacteria are located at the front of the group where the gradient is shallower; and cells that are less sensitive are located towards the back where the gradient is steeper.

These findings suggest a general principle for how bacteria form ordered patterns whilst migrating as a collective group. This behavior could also apply to other populations of distinct individuals, such as ants following a trail or flocks of birds migrating in between seasons.

different phenotypes determine their location in the group is an essential prerequisite to uncover the organization principles of collective populations.

The chemotactic microbe, *Escherichia coli*, provides a simple model to address the emergence of collective decision-making among diverse population, as it can exhibit both individualistic behaviors (*Dufour et al., 2016*; *Frankel et al., 2014*; *Kussell and Leibler, 2005*; *Waite et al., 2016*; *Waite et al., 2018*) and collective migratory patterns (*Adler, 1966a*; *Fu et al., 2018*; *Keller and Segel, 1971a*, *Wolfe and Berg, 1989*). Individual cells perform run-and-tumble random motions by spontaneously switching the rotating direction of flagella (*Berg, 2004*; *Berg and Brown, 1972*). These cells can facilitate the chemotaxis pathway to control the switching frequency to bias their motions toward their favorable direction along the chemoattractant gradient, where the efficiency to climb the gradient is defined as the chemotactic ability ($\chi$) (*Celani and Vergassola, 2010*; *Dufour et al., 2014*; *de Gennes, 2004*; *Si et al., 2012*). In addition, the chemotactic abilities of individual cells exhibit substantial phenotypic heterogeneity even for the clonal bacterial population, which diversifies the chemotactic response to identical signals (*Spudich and Koshland, 1976*; *Waite et al., 2016*; *Waite et al., 2018*). Despite the stochastic solitary behavior and variations in phenotypic ability, the *E. coli* population can migrate as a coherent group by following a self-generated attractant gradient, via consumption of the whole population (*Adler, 1966a*; *Saragosti et al., 2011*; *Wolfe and Berg, 1989*). The migratory group form a stable pattern of phenotypes sorted by their chemotactic abilities (*Figure 1A*), which is believed to maintain phenotypic diversity in the group (*Fu et al., 2018*; *Waite et al., 2018*). Although a previous study showed that behavior modulation helps migrating bacteria to maintain a consistent group (*Saragosti et al., 2011*), how individuals

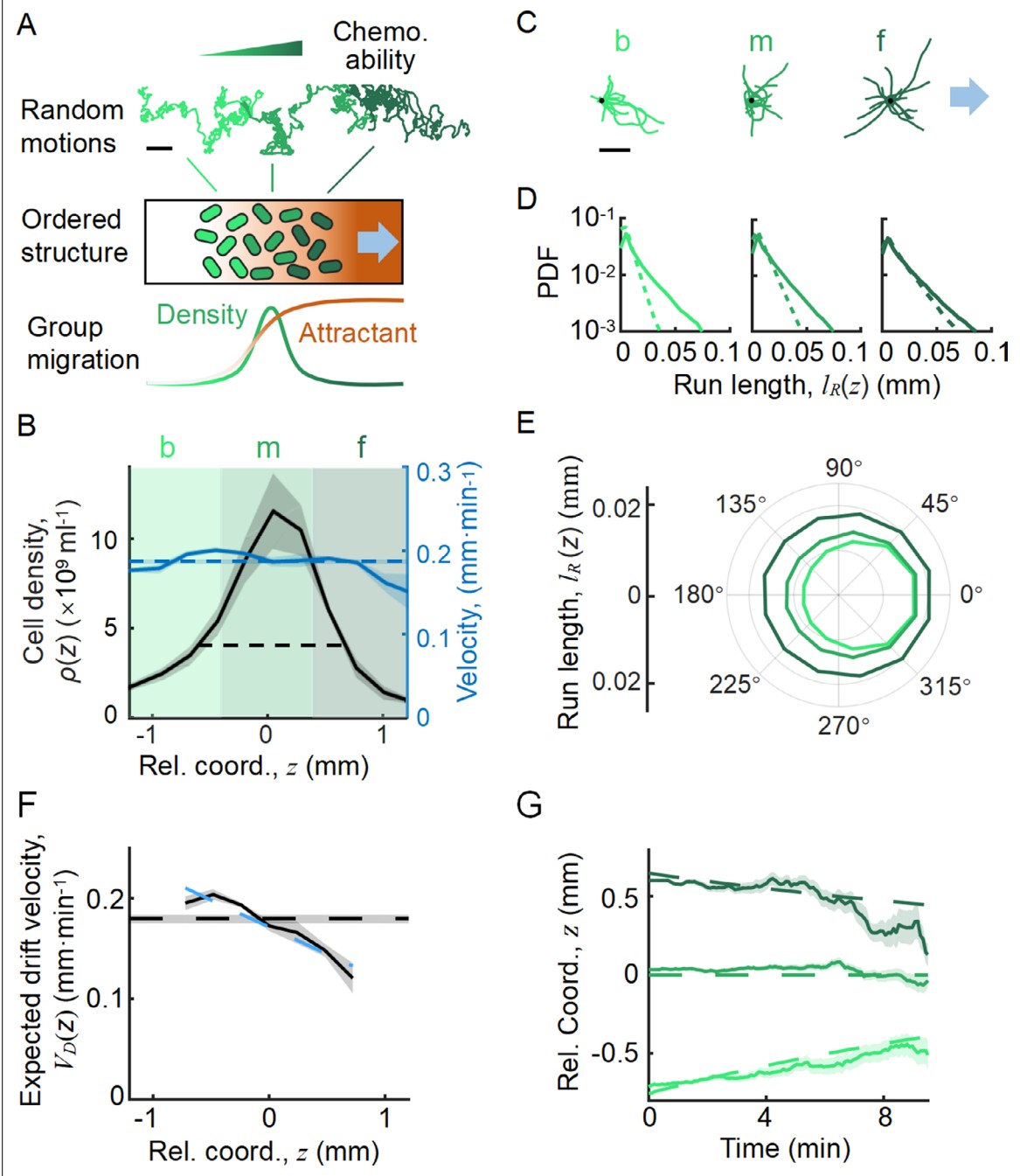

**Figure 1.** Statistics of single-cell behavior during collective group migration. (**A**) Bacterial population of diverse phenotypes sorted according to their chemotactic abilities (increasing from light green to dark green) during collective migration following the self-generated attractant gradient (brown color). Meanwhile, individual cells perform run-and-tumble random motions biased toward the migration direction. Scale bar reflects 0.1 mm. (**B**) Bacterial density profile $\rho(z)$ is stable (black solid line) in the moving coordinate $z = x - V_G t$. Where the width, represented by the black dashed-dotted line, is defined by two times the standard deviation of the bacterial relative position ($2\sigma$, black dashed line). The instantaneous velocity ($V_I(z)$) (blue solid line) is uniform and is equal to the average group velocity $V_G$ (blue dashed line). This observation was independent of the sampling time interval (*Appendix 1—figure 2*). (**C**) Sample runs of bacteria aligned by their initial positions (black dots) from the three regions (defined by colored regions in **B**) showed that cells at the back of the group tended to run forward, compared with cells in other regions. b, m, and f stand for the back, middle, and front of the migration group, respectively. Scale bar reflects 5 µm. (**D**) The exponential distributions of forward runs (solid lines) and backward runs (dashed lines) suggested the increasing efficiency of runs from the back to the front. (**E**) The mean run length in different directions confirmed the increasing efficiency of runs. (**F**) The expected drift velocity $V_D(z) = \frac{\langle l_R(z) \cdot \cos\theta_R(z) \rangle}{\langle \tau_R(z) + \tau_T(z) \rangle}$ (black solid line) decreased from the back to the front, and is linearly fitted by $V_D(z) \approx -rz + V_{D0}$, with $r = 0.05 \text{ min}^{-1}$ and $V_{D0} = 0.17 \text{ mm} \cdot \text{min}^{-1}$. $V_D(z)$ crossing with the average group velocity $V_G$ (black

*Figure 1 continued on next page*

*Figure 1 continued*

dashed line), implied that bacteria perform mean reversion motions. The $V_D(z)$ profile was cut to present the majority of cells (~90%) ($\pm 1.65\sigma$). (**G**) The time evolution of the average position ($z$, solid lines) of cells starting from the back, middle, and front of the migration group (color code was defined in **B**) confirmed the reversion behavior. Shaded area represents the s.e.m. of more than 450 cells (see Materials and methods). Analytically, the OU-type model predicted that $z = C_0 e^{-rt} - (V_{D0} - V_G)/r$ (dashed lines), where $C_0$ can be fitted by the starting position (see supplementary text). In panels **B**, **F**, and **G**, the shaded area represents the s.e.m. of three biological replicates. The spatial bin size was 240 μm .

with phenotypic variations manage to determine their relative positions in the group remains to be determined.

Here, we analyzed single-cell trajectories of bacterial run-and-tumble motions in the chemotactic migration group (see Materials and methods). We found that the expected drift velocity of individual cells decreased from the back to the front. Such a spatial profile modulates cells to behave as mean reverting processes relative to the entire group, that is, cells effectively tend to revert their direction of runs toward the mean position of the group. Using an Ornstein-Uhlenbeck (OU)-type model, we demonstrated that the mean reversion behavior is a result of a pushed wave, where the driving force decreases from the back to the front of the group. Cells of different phenotypes are imposed to the same type of driving force, of which the strength is coupled with their chemotactic abilities. As a result, the pushed wave front, driven by the spatially structured force, can maintain more diverse individuals in the migratory group. By theoretical analysis and stochastic simulations, we also discovered that the balanced locations of diverse phenotypes are spatially ordered by their chemotaxis abilities. Further simulations and experiments with cells of titrated chemoreceptor abundance demonstrated that this spatial modulation of individual behavior enables the ordering of bacteria with diverse chemotaxis abilities. Therefore, although the high-order computational abilities are not available to simple organisms, the spatial modulation of stochastic behaviors at the individual level reveals novel decision-making capabilities at the population level.

## Results

### The drift velocity of individual cells during group migration exhibits a spatially structured profile

To directly investigate the ordering mechanism in a coherent migration group, we quantified the stochastic behaviors of bacterial run-and-tumble motions relative to the stable migrating group. To achieve this, we employed a microfluidic device that generated a stable propagating band of bacteria, as previously reported (*Fu et al., 2018*; *Saragosti et al., 2011*). Using aspartate (Asp) as the only chemoattractant to drive the migration of *E. coli*, we tracked a small fraction of fluorescent bacteria (JCY1) as representatives of the non-fluorescent wild-type cells (RP437) (see Materials and methods, *Appendix 1—figure 1*, *Video 1*). Because the group velocity $V_G = \langle \Delta x_i(t)/\Delta t \rangle$ was constant over time, $V_G \sim 3.0$ μm/s (*Appendix 1—figure 2*), we were able to map the tracks to a moving coordinate $z = x - V_G t$.

With the shifted tracks, we calculated the key statistics of the single-cell behaviors. We first checked that the average instantaneous velocity, $V_I(z) = \langle \Delta x_i(z)/\Delta t \rangle$, was constant along the density profile $\rho(z)$ (*Figure 1B*). This result confirmed that the group migrates coherently. Then, we identified all the run states and tumble states of individual trajectories using a previously described computer assistant program (*Dufour et al., 2016*; *Waite et al., 2016*), to ensure that the tracks are separated into successive tumble-and-run events (see Materials and methods). Comparing the sample events initiated from the back (b), middle (m), and front (f) of the migration group, we observed that the runs in the front were longer but distributed more uniformly in terms of the directionality, whereas the runs in the back were shorter but more oriented toward the group migration direction (*Figure 1C*, *Appendix 1—figure 3A, B*).

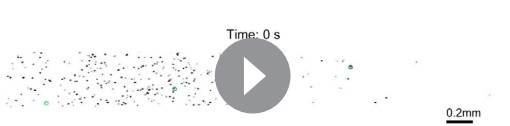

Time: 0 s

0.2mm

**Video 1.** Bacterial tracking in the migration group. Each dot represents a bacterium captured under microscope. Lines represent typical tracks. The colors of tracks from light green to dark green represent the mean positions of the tracks from the back to the front of the migrating group.

https://elifesciences.org/articles/67316/figures#video1

Quantitatively, the statistics of run length (and duration) displayed exponential distributions with the means for the direction of group migration longer than those for the opposite direction (*Figure 1D*, *Appendix 1—figure 3C*). The angular distribution of the run length confirmed the difference in directionality between cells in different spatial locations (*Figure 1E*, *Appendix 1—figure 3D*). We also confirmed that cells in the back showed greater directional persistence toward the migration direction (*Appendix 1—figure 4A–C*), as (*Saragosti et al., 2011*) reported previously. All these results suggested that the bacteria in the back run more effectively toward the direction of group migration than those in the front.

To quantify the spatial extent of the drift efficiency, we defined the expected drift velocity $V_D \equiv \frac{\langle l_R \cdot \cos \theta_R \rangle}{\langle \tau_R + \tau_T \rangle}$ , by the projection of the average run length along the migration direction over the average duration of runs and tumbles (*Dufour et al., 2014*). This quantity reflects the effective run speed of run-and-tumble events that start running on a given location relative to the group. The drift velocity was found to decrease from the back to the front, crossing the group velocity $V_G$ in the middle of the group (*Figure 1F*). This particular trend of $V_D(z)$ suggests a mean reverting behavior of bacteria in the group: the cells at the back drift faster than the group ($V_D > V_G$), enabling the cells to catch up with the group; at the same time, the cells in the front drift slower than the group ($V_D < V_G$), causing them to slow down and fall back (*Figure 1G*). Such mean reverting process also results in sub-diffusion of individuals relative to the group (*Appendix 1—figure 3F*), for which the mean square displacement (MSD) is constrained over time (*Appendix 1—figure 3G*). The slope of the spatial extent of $V_D(z)$, $-r = -0.05 \ \mathrm{min}^{-1}$ , quantifies the speed at which individuals revert to its center.

We noted that the spatial profile of the expected drift velocity $V_D(z)$ was different from the instantaneous velocity $V_I(z)$ . This is because the instantaneous velocity $V_I(z)$ defines the average speed of cells in a given time interval $dt$, which reflects the positional shift of a group of cells at a given position. However, the expected drift velocity $V_D(z)$ defines the average speed of run-and-tumble events that start tumbling at a given position. Since the run duration is explicitly modulated by the gradient of chemoattractant $g(z)$ and is dependent on the chemotactic ability $\chi$ , $V_D(z) = \chi g(z)$ represents the drift velocity driven by the external stimuli (*Celani and Vergassola, 2010*; *de Gennes, 2004*; *Dufour et al., 2014*; *Si et al., 2012*).

## The decreasing drift velocity implies a pushed wave front of group migration

To understand how the spatially structured profile of the drift velocity $V_D$ impacts on the group migration, we first adopted a Langevin-type model that describes bacterial motions as an active Brownian particle driven by the expected drift velocity $V_D$ and a random force: $dx = V_D dt + \epsilon dW$ (*Berg, 2004*). In this model, the run-and-tumble random motions are considered as a Gaussian random force $\epsilon dW$, while the cell motions are imposed to a deterministic force $V_D$ (*Rosen, 1973*; *Rosen, 1974*). The strength of Gaussian noise can be estimated by the effective diffusion coefficient $\epsilon = \sqrt{2D}$ , while the drift velocity is determined by the product of the perceived gradient $g(z)$ and the chemotactic ability $\chi$, $V_D = \chi g(z)$ . Such a stochastic description of bacterial motions has been proven equivalent to the classic Keller-Segel (KS) model that described the population dynamics of bacterial chemotactic group migration (*Keller and Segel, 1971a*, *Rosen, 1973*). In the moving coordinate, $z = x - V_G t$ , this Langevin-type model specifies that $dz = V_D(z) dt - V_G dt + \epsilon dW$. Thereby, the cell motions relative to the migrating group are modulated by two effective forces: one generated by $V_D(z)$, which pushes the cells to catch up with the wave; and another generated by $-V_G$ , which drags the cells to fall behind the wave. These two 'forces' constrain the random motions of individuals in an effective potential well $U(z)$ (*Figure 2A*).

To estimate the spatial profile of the effective driven force $V_D(z)$, we analyzed the KS model with moving ansatz (supplementary text). Assuming the density profile $\rho(z)$ directly measured from experiments (*Figure 1B*), we can deduce the chemoattractant concentration profile $S(z)$ (*Appendix 1—figure 4D*), as well as the perceived gradient $g(z)$ (*Figure 2B*). Since the perceived gradient $g(z)$ is almost linear in the main part of the wave profile, we approximated it by $g(z) \approx g_0 + g_1 z$ (*Figure 2B*, dashed line), which allows us to transform the stochastic model to an OU-type equation:

$$dz = \chi g_1 z dt + (\chi g_0 - V_G) dt + \epsilon dW \tag{1}$$

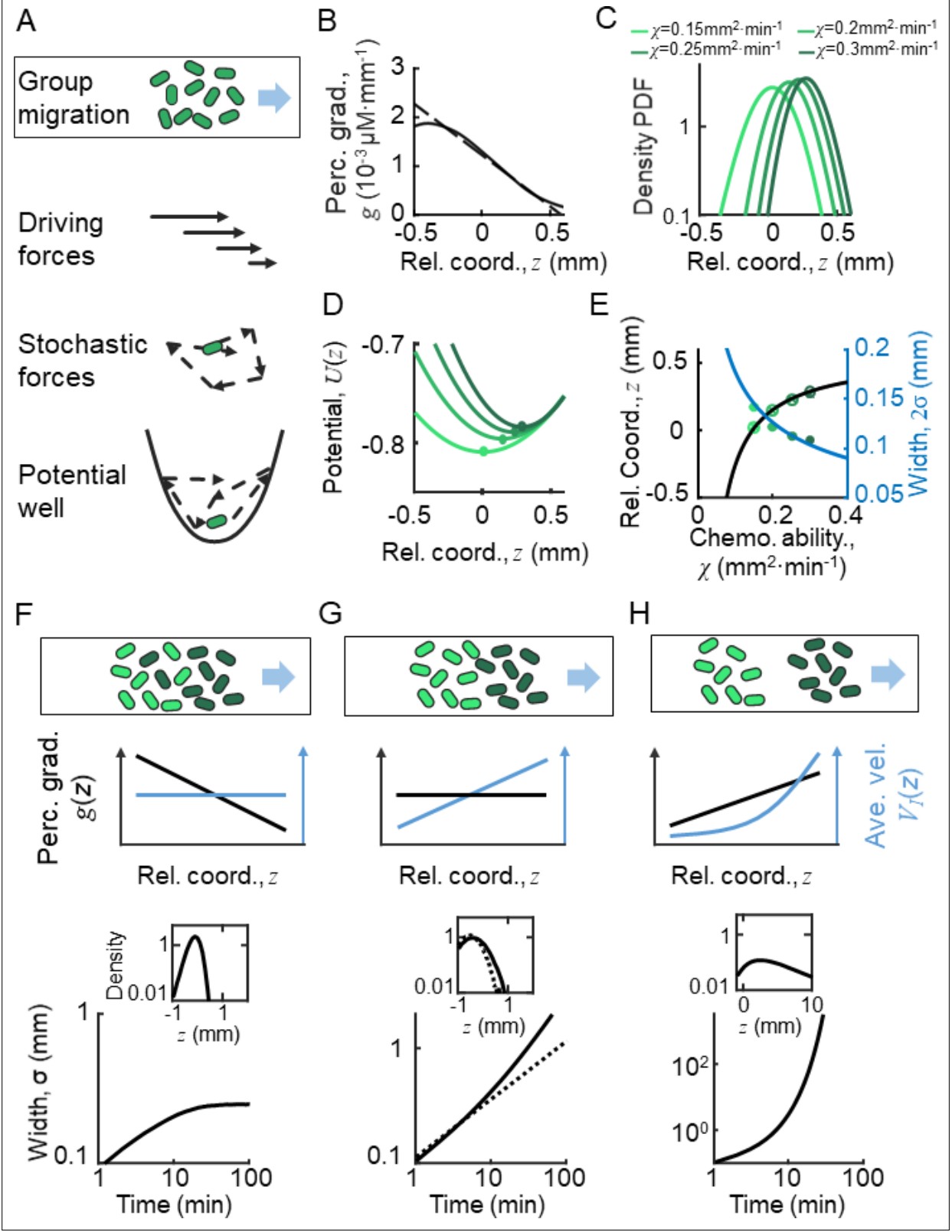

**Figure 2.** Ordered location of phenotypes of active particles in a moving gradient as a result of a pushed wave. (**A**) Illustration of the Langevin-type model where the migrating bacteria are modeled by active particles under a deterministic decreasing driving force and a stochastic force. These forces allow the particles to form collective migration. On the moving coordinate of the migration group, active particles are bounded in a potential well generated by the driving force and the motion relative to the moving group ($-V_G$). (**B**) The perceived gradient $g(z)$ (blue line) is deduced from $S(z)$,

*Figure 2 continued on next page*

*Figure 2 continued*

which is further calculated from the experimentally measured density profile (*Appendix 1—figure 4D*). This gradient profile was then transformed to a moving gradient $g(x,t) = g(z) + V_G t$, and applied in the simulations of different active particles following Langevin dynamics. A linear fit of $g(z)$ around $z = 0$ is plotted (dashed blue line) and is applied to the Ornstein-Uhlenbeck (OU)-type model. (**C**) Particles migrate collectively with a moving gradient, while they are located in ordered locations according to their phenotype $\chi_i$ (green lines). Colors from light to dark represent increasing $\chi$. (**D**) The effective potential wells generated by the effective force ($\chi g(z)$) are spatially ordered. Circles present the minimum of the potential wells. (**E**) The peak positions (cross) and the mean positions (circles) increase with $\chi$. These were predicted by the OU-type model (Equation S15) (black line). The width of the density profile is defined by two times the standard deviation of all cells ($2\sigma$), which decreases with $\chi$, which was consistent with the prediction of Equation S18 (blue line). (**F–H**) Group migration of mixed phenotypes under decreasing, invariant, and increasing gradient. Simulations were performed in one dimension (1D) with the chemotaxis ability of the group following a Gaussian distribution with a mean of 0.11 mm² min⁻¹ and a variation of 0.02 mm² min⁻¹ (*Appendix 1—figure 5C*). The force field used in the simulation followed the linearized force field, as in Equation 1, while $g_1 > 0$ in F, $g_1 = 0$ in G, and $g_1 < 0$ in H. All force fields were set as moving with velocity $V_G$. The dashed line in **G** represents a group with a single phenotype of $\chi = 0.11$ mm²min⁻¹. Insert plots represent the wave front in the moving coordinate after 10 min of simulation. The width of the moving group, defined by the standard deviation of all particles, converges under decreasing gradient (**F**) while it diverges in cases of **G** and **H**.

By this simplified model, we obtained a clear picture how individual behaviors are regulated relative to the group: cells are imposed to a driving force linearly dependent on their relative positions in the group, $F(z) = \chi g_1 z + (\chi g_0 - V_G)$. This suggests that the random motions of bacteria are constrained in a parabolic moving potential well, $U(z) = \frac{1}{2}\chi|g_1|z^2 + (\chi g_0 - V_G)z + U_0$ (*Figure 2A*), where $U_0$ is set by $U(\infty) = 0$. The position that minimizes $U(z)$ is also the balanced position, $z_0 = -\frac{g_0}{g_1} - \frac{V_G}{\chi|g_1|}$, where the driving force is null $F(z_0) = 0$. Behind the balanced position, cells experience a pushed force to catch up the group. Cells would start to fall back once they exceed the balanced position, where the driving force becomes negative. Therefore, the decreasing profile of the driving force enable cells to perform mean reverting behaviors around the balanced position. In addition, the expected rate that cells tend to revert back to the balanced position is defined by the slope of the spatially dependent driving force, $r = \chi|g_1|$ (*Figure 2B*).

Given the knowledge of individual behaviors, we studied the spatial distribution of population on the group migration. The OU model (Equation 1) which describes the single-cell stochastic motions has an equivalent form, known as the Fokker-Planck equation (Equation S19) which describes the spatial-temporal dynamics of cell density distribution. This population model provides a traveling wave solution with the mean position around $z_0$ and standard deviation $\sigma = \frac{\epsilon}{\sqrt{2\chi|g_1|}}$.

From the solution, we noted that the effective driving force, as well as the drift velocity, has a negative slope ($g_1 < 0$). The decreasing profile of the drift velocity suggests that the leading front of the group migrates as a pushed wave front (*Gandhi et al., 2016*; *van Saarloos, 2003*). As a key feature of the pushed wave, the leading front of the group drops parabolically, which is much faster than that of diffusion front. This further leads to a tight density profile of group migration for a single phenotype population.

## The pushed wave front results ordered pattern of phenotypes

To address whether this pushed wave front still holds in presence of phenotypic diversity, we further examined the above analysis with cells of diverse chemotactic abilities $\chi_i$ imposed to the same moving perceived gradient $g(z)$ (*Figure 2B*). Given a monotonically decreasing profile of perceived gradient $g(z)$, the driving force that each phenotypic individual experiences exhibits the same spatial dependency with the slopes depend on the intrinsic phenotypic properties of each phenotype $r_i = \chi_i|g_1|$. This monotonic dependency means that the balanced positions $z_0$ of the diverse phenotypes are orderly arranged. By the stochastic Langevin-type model with phenotypic diversity in chemotactic ability, we first confirmed that each phenotypic population migrates at a constant speed $V_G$, following the moving gradient $g(z)$ (see supplementary text). The density profiles of cells with different $\chi_i$ follow the same shape but are spatially orderly aligned (*Figure 2C*). Under the same moving gradient $g(z)$, the driving force $\chi_i g(z)$ is phenotype-dependent, so that the bottom position of the potential well, $z_{0,i} = -\frac{g_0}{g_1} - \frac{V_G}{\chi_i|g_1|}$, is also spatially arranged according to $\chi_i$ (*Figure 2D*). As predicted by the OU model, the balanced positions $z_{0,i}$ of different phenotypes increase with their chemotactic abilities $\chi_i$ (*Figure 2E*, black line), while the standard deviation ($\sigma_i$) of the density profiles decreases with $\chi_i$ (*Figure 2E*, blue line). We also confirmed the ordered structure of phenotypes by a particle-based model of the Langevin-type dynamics coupled with chemoattractant consumption

(*Appendix 1—figure 5*). Therefore, under the spatially decreasing driving force, cells with phenotypic diversity perform the same type of mean reverting processes with spatially ordered mean positions.

The spatial order of phenotypes does not directly promise a compact group migration with phenotypic diversity. By close examination of the density profile, we found that each phenotypic subpopulation propagates as a pushed wave front. We further calculated the total density profile of the entire migratory group with Gaussian distribution of chemotactic abilities under a decreasing linear gradient. Simulation reveals a pushed wave front for the combination of these subpopulations with different chemotactic abilities (*Figure 2F*, insert). In addition, we checked that the width of the entire group maintains in a converged width over long time, suggesting that the pushed wave profile enables the migratory population with diversity to keep in a tight shape (*Figure 2F*).

We examined the migration profiles under other forms of perceived gradient $g(z)$: a constant perceived gradient and a spatially increasing perceived gradient. In the first case, individual bacteria of identical phenotype follow the diffusion process relative to the group, where the standard deviation of the population increasing with time by $\sigma = 2\sqrt{Dt}$ (*Figure 2G*, dashed line). Each phenotype subpopulation is expected to have a constant drift velocity over space. However, as the drift velocity would vary by the chemotactic ability, each subpopulation migrates in different group speeds, suggesting a compact group migration of diverse population cannot be maintained in a constant perceived gradient (*Figure 2G*). In the latter case, when imposed to a spatially increasing driving force (a pulled wave, by definition), individual cells display super-diffusion processes. The density profile easily diverges in this case (*Figure 2H*). Therefore, we concluded that the pushed wave front, driven by a decreasing shape of driving force, enables a spatially ordered and compact pattern of phenotypes while on the move.

## Spatially ordered individual behaviors predicted by agent-based simulation

Although the simplified OU-type model (Equation1) represents a key aspect of the ordering mechanism of phenotypes, it does not detail the signaling processes of bacterial chemotaxis, such as receptor amplification, adaptation, and motor responses (*Sumpter, 2010*), and it cannot predict bacterial run-and-tumble behavior. To consolidate the proposed mechanism underlying the emergence of spatial orders from the individual random motions, we further performed agent-based simulations integrated with the chemotactic pathway, multi-flagella competition, and boundary effect in three dimensions (3D) (*Dufour et al., 2014*; *Jiang et al., 2010*; *Sneddon et al., 2012*). In the agent-based model, the attractant dynamics governed by diffusion and bacterial consumption is described by a reaction-diffusion equation (Equation S2) as previous multiscale models (*Erban and Othmer, 2005*; *Xue and Othmer, 2009*). We constructed a population with multiple phenotypes defined by different chemotactic abilities $\chi_i$, where $\chi_i$ was varied by changing the receptor gain $N$ (for details, see supplementary text). Since the receptor gain $N$ only affects the amplification factor by which a cell responds to the gradient, the variation in bacterial motility $\epsilon$ is unchanged. As a result, a dense band of migrating cells that follow a self-generated moving chemoattractant gradient via consumption were recaptured as experiments (*Appendix 1—figure 6A*). The phenotypes were spatially ordered as $\chi$ varies (*Appendix 1—figure 6B*), and the velocity profile of each phenotype decreases from the back to front (*Appendix 1—figure 6C*) as predicted by the OU-type model.

To better analyze the simulations, we simplified the simulation with a non-consumable attractant profile $S(z)$ moving with constant speed $V_G$ (*Appendix 1—figure 4D*). Using this simplified model, we first checked that the mean positions of the density profiles of cells with different receptor gains $N$, as well as their peaks, were orderly aligned with respect to chemotactic ability $\chi_i$.

As an important advantage of the agent-based simulations, the model allowed us to analyze single-cell behavior during the ordered group migrations. For each phenotype , the expected drift velocity $V_{D,i}(z)$ decreased along the density profile (*Figure 3A*). Consistent with the ordered structure of the density profiles, the intersection between $V_{D,i}(z)$ and $V_G$ exhibited the same order of chemotactic ability $\chi_i$ (*Appendix 1—figure 7*). As the reversion rate $r_i = \left| \frac{dV_{D,i}(z)}{dz} \right|$ showed a positive correlation to $\chi_i$, cells with lower receptor gain $N$ (resulting in a smaller $\chi$) experienced a weaker reverting force toward the center (*Figure 3A* insert). Thus, the effective moving potential, $U_i(z)$, which constrains the cells around the mean positions sorted by their chemotactic abilities, becomes flat for cells with lower chemotaxis ability $\chi$ (*Figure 3B*; *Long, 2019*). As a result, cells of each

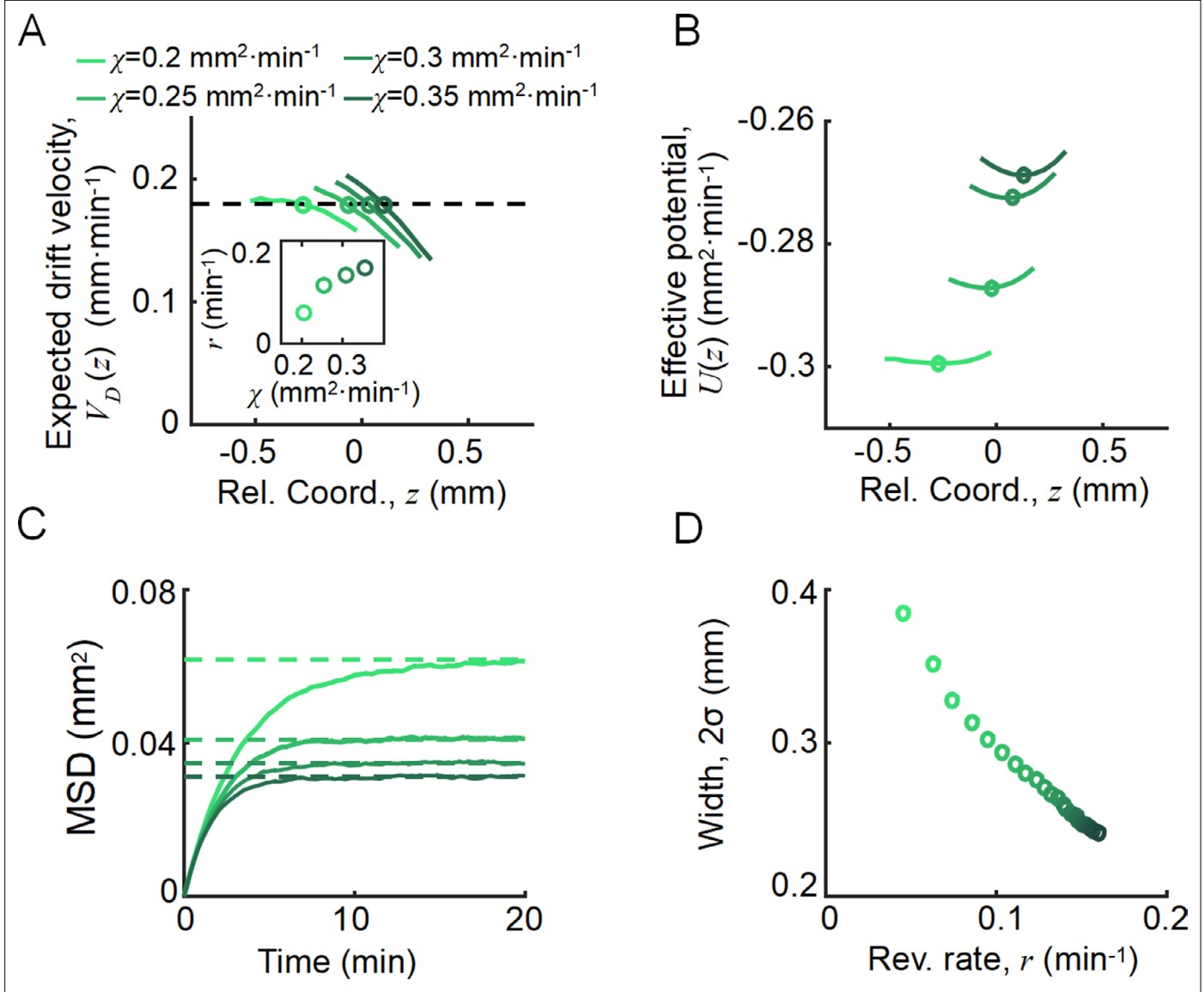

**Figure 3.** Agent-based simulations recapture the ordered behavior of individuals. (**A**) The expected drift velocity $V_D(z)$ of simulated bacteria decreased from the back to the front of the migration group, where the chemotactic ability $\chi$ ranged from 0.2 to 0.35 mm²min⁻¹ , which was consistent with the experimental results shown in **Figure 1F**. The intersections between the $V_D$ curves with the preset group velocity $V_G$ (black dashed line) shifted toward the back of the migration group as $\chi$ decreased (circles). The different colors of the lines and circles correspond to different chemotactic abilities $\chi$, as shown in the legend. The same color coding also applies to (**B–D**). (**B**) The reversion rate $r_i = |dV_{D,i}(z)|/dz|$ increased with chemotactic ability. (**C**) The effective potential well calculated by $U_i(z) = \int_z^{+\infty} V_{d,i}(z)\,dz$. Positions of the potential minimum $z_{min}$ are marked as circles. As illustrated, for a lower chemotaxis ability $\chi$, the potential well is shallower and $z_{min}$ shifts toward the back of the migration group. (**D**) The width of the density profile (measured by $2\sigma$, see **Figure 1B**) decreases with the reversion rate $r_i$ as well as the chemotaxis ability $\chi_i$ . The mean square displacement (MSD) of bacteria (insert, solid lines) is bound to $2\sigma_i^2$ (insert, dashed lines) (see supplementary text). In panel (**A, C**), curves were cut to present the majority of cells (90%) ($z_{min} \pm 1.65\sigma_i$). More details on the results of this simulation are presented in **Appendix 1—figure 7**.

phenotype perform sub-diffusion, whereby the MSD along the migration coordinate relative to the group was bound at a level negatively correlated to $\chi$ (**Figure 3C**). The width of the density distribution, as an effect of the reversion force, decreased with the reversion rate, as an approximate linear function. Using this agent-based model, we further obtained similar results for populations of different $\chi_i$ through adaptation time $\tau$, or basal CheY protein level $Y_{p0}$ , which determined the basic tumble bias $TB_0$ (**Dufour et al., 2014**; **Jiang et al., 2010**; **Sneddon et al., 2012**; **Appendix 1— figure 8**).

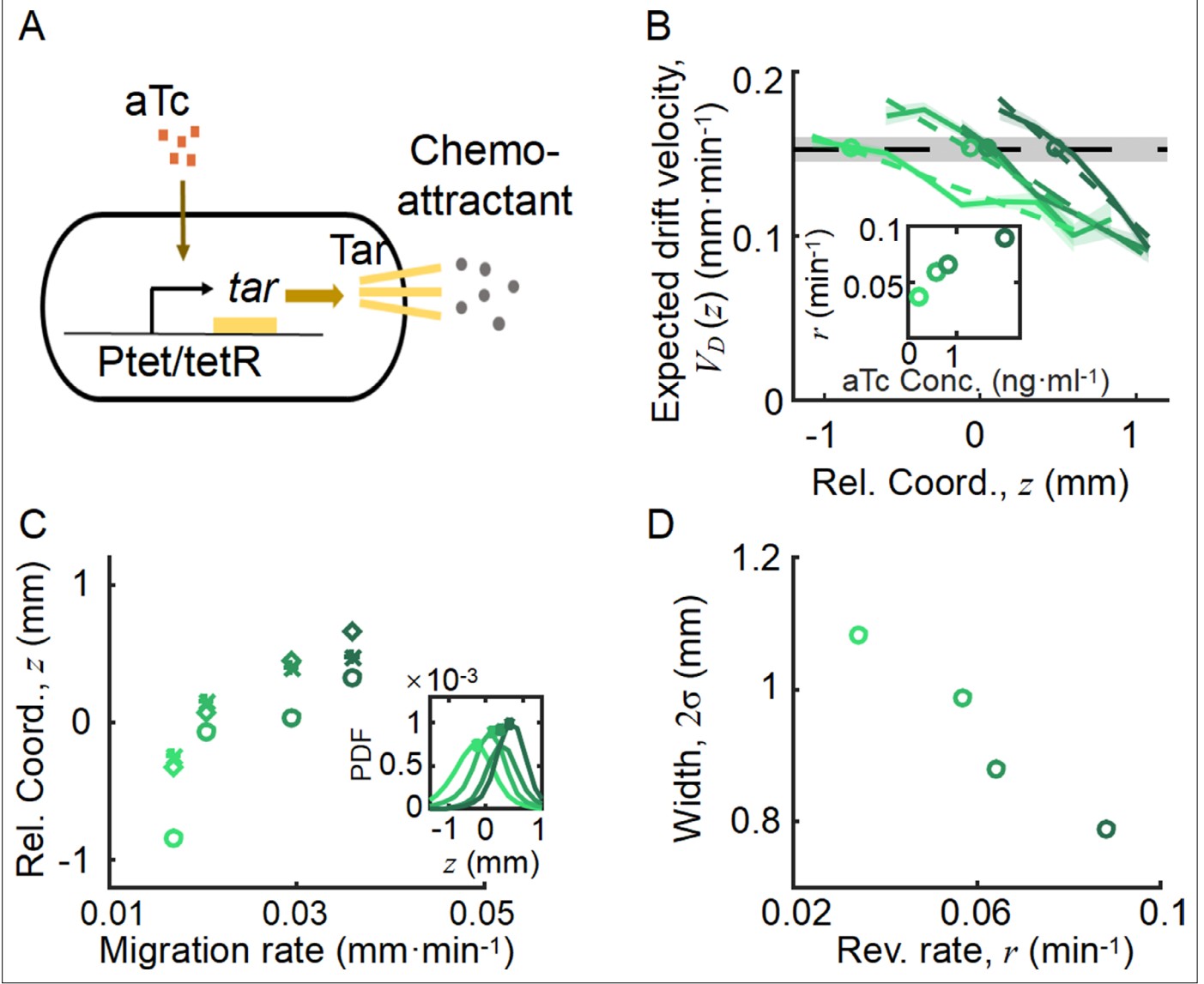

**Figure 4.** Spatial ordered structure emerged from the behavioral modulation of cells with different chemoreceptors. (**A**) Genetic circuit of the Tar-titratable strain. In the experiments, the expression level of Tar (a chemoattractant receptor protein) was titrated by the concentration of external inducer (aTc). The chemotactic ability $\chi$ of bacteria was then determined by the expression level of Tar (**Adler, 1969**). (**B**) The expected drift velocity $V_{D,i}(z)$ of the Tar-titratable strain JCY20 (colored solid line) was spatially modulated and decreased from the back to the front of the migration group and intersected with a group migration velocity $V_G \approx 0.15$ mm/min (black dashed line). The linear fit of $V_{D,i}(z)$ (colored dashed lines) intersected with $V_G$ at positions (circles) determined by the corresponding Tar expression level. Colors from dark to light green corresponded to inducer (aTc) concentrations of $[1, 3, 6, 20]$ ng/mL. The black shaded area of $V_G$ represents the s.d. of four experiments, while the colored shaded area of the $V_D$ curves presents the s.e.m. of the counted runs. (**C**) In the experiment, the positions of the $V_{D,i}(z) - V_G$ intersections (circles, illustrated in B), together with the peaks (stars, illustrated in the insert figure), and the average positions (diamond) of the bacterial density profiles, all shifted toward the front of the migration group for strains with higher Tar expression levels, which had higher chemotactic abilities and migrated faster on agar plates (x-axis, see Materials and methods; *Appendix 1—figure 9*). The related density profiles (PDF) were shown in the insert plot and the color coding of lines/symbols in both panels C and D was the same as that in B. (**D**) The width of density profiles ($2\sigma$) of Tar-titrated bacteria decreased with the reversion rate $r$.

## Experiments with titrated phenotypes confirm ordered behavior modulation

To verify the model predictions on the individual behaviors of different phenotypes, we experimentally measured the trajectories of cells with different chemotactic abilities during the group migration. Specifically, we altered the chemotaxis abilities of cells by titrating the expression level of Tar, which is under the control of a small molecule inducer aTc (*Sourjik and Berg, 2004*; *Zheng et al., 2016*) (see

Materials and methods and *Figure 4A*). The variations in the expression of Tar would lead to different receptor gains in response to the Asp gradient (*Adler, 1966b*; *Adler, 1969*; *Sumpter, 2010*), but the tumble bias and growth rate would not change. Using the migration speed of the bacterial range expansion to quantify the chemotaxis ability of the titrated strains, we found that the chemotaxis ability increases with aTc concentration (see Materials and methods and *Appendix 1—figure 9*).

The Tar-titrated cells labeled with yellow fluorescent protein (YFP; strain JCY20) were added to the wild-type population at a ratio of 1 in 400. Within the wild-type population, 1 in 50 cells was labeled with red fluorescent protein (RFP) (strain JCY2). As the Tar-titrated strain constituted a small portion of the pre-mixed population, we considered that the density profile of the population would be invariant to different levels of induction of Tar. The premixed population could generate collective group migration, similar to the wild-type population (*Appendix 1—figure 9*). The trajectories of the YFP-labeled cells were tracked to represent the behavior of cells with different chemotactic abilities, while the profile of wild-type cells with RFP was also measured to characterize the density distribution of the entire migratory population.

By comparing the statistics of cells with different Tar expression levels, we found that the expected drift velocity $V_{D,i}\left(z\right)$ followed the same decreasing pattern from back to front (*Figure 4B*). More importantly, as the Tar expression level (chemotactic ability) increased, the slope of the decreasing pattern increased, which was consistent with the model prediction shown in *Figure 3A*. The intersections between $V_{D,i}\left(z\right)$ and $V_G$, as well as the peak positions and mean positions of each Tar-titrated density profile (*Figure 4C*), shifted toward the front as the chemotactic ability increased (as measured by the migration rate on agar plates; *Cremer et al., 2019*; *Liu et al., 2019*). The $V_D$ cross point was always behind the peak position and the mean position (*Figure 4C*), suggesting that cells were leaking behind. Moreover, the width of each Tar-titrated density profile (defined by $2\sigma_i$) decreased as the reversion rate $r_i$ increased (*Figure 4D*), consistent with the model results in *Figure 3C*. Thus, as the OU-type model predicts, the width of the density profile is controlled by the reversion rate determined by the chemotactic ability $\chi_i$.

## Discussion

In summary, coordinated behaviors with ordered spatial arrangements of phenotypes are abundant in a wide range of biological and human-engineered systems, and are believed to involve elaborate control mechanisms. For animal migrations, it is challenging to characterize simultaneously the computational strategy and behavior at the individual level so as to avoid averaging out phenotypic diversity, and the emergent behavior at the population level (*Couzin et al., 2005*; *Couzin et al., 2002*; *Vicsek and Zafeiris, 2012*). For bacterial chemotactic migration, cells with different phenotypes are spatially aligned based on their chemotactic abilities. This observation was explained as a self-consistent result with the decreasing profile of attractant predicted by KS model (*Fu et al., 2018*). In this study, we demonstrated that the collective consumption of attractant by bacterial group generates a spatial structure of individual drift velocity along the migrating group profile. Such a spatial profile of drift velocity results a pushed wave front on population level, and maintains diverse phenotypes in a compact migration group. Moreover, this pushed wave front enables spatial modulation of individuals to perform mean reverting random motions around centers sequentially aligned by their chemotactic abilities, thereby giving rise to a spatially ordered pattern. Therefore, we demonstrated that the population order could emerge among diverse individuals that following the same rule of behavioral modulation.

This strategy of self-organization does not require sophisticated communications (*Curatolo et al., 2020*; *Karig et al., 2018*; *Liu et al., 2011*; *Payne et al., 2013*) nor other hydrodynamic interactions (*Chen et al., 2017*; *Drescher et al., 2011*; *Zhang et al., 2010*) among individuals. Our observation of the decreasing drift velocity can imply the effective perceived gradient that cells experience. We believe that this spatial profile is mainly contributed by the consumption of the chemoattractant. By using a Tar-only strain (UU1624) (*Gosink et al., 2006*), we demonstrated that the mutant could also generate a stable group migration in our experimental condition similar to the wild-type strain (*Appendix 1—figure 9G*). This further suggests that the secretion of self-attractants is unlikely a necessary condition of collective group migration (*Cremer et al., 2019*; *Fu et al., 2018*; *Keller and Segel, 1971a*), although there are doubts about the existence of self-attractants in high density (*Budrene and Berg, 1995*).

The spatially dependent drift velocity provides a structured driving force of a migration group, resulting a pushed wave front. Pushed and pulled waves are determined by the spatial distribution of the spreading velocity of a propagation front (*van Saarloos, 2003*; *Figure 2F–H*). The properties of pushed and pulled waves have been discussed in growth-driven range expansion (*Birzu et al., 2018*; *Erm and Phillips, 2020*; *Gandhi et al., 2016*), where the wave type is determined by density-dependent growth rates. A prominent example of pulled wave is known as the Fisher wave (*Fisher, 1937*), where the population expansion is driven by constant diffusion and logistic growth of individuals. However, in such biological systems, the spatial dependence of front speeds is hardly quantified in experiments. Here, by direct measurement of drift velocity on single-cell level, we identified the chemotactic migration group of bacteria as a pushed wave. This chemotaxis system would provide a unique multi-scale model to study further details of pushed wave.

The spatially decreasing profile of driving force does not only cause an ordered pattern of phenotypes, but also results a pushed wave front that enables a negative feedback control on the propagating speed. This further allows a compact density profile for heterogenous population to migrate at the collective level. The advantageous to keep diversity in the pushed wave front was also reported in the growth-driven range expansion system (*Birzu et al., 2018*). Our study revealed that, other than spatial regulation of fitness, the direct modulation of individual drift velocity in space could also maintain diversity in range expansion.

Detailed analysis of the spatial structured driving force could also provide the limits of phenotype that is allowed in the group. In the migratory group, the same rule of behavioral modulation applied to cells with different phenotypes, such that the random motions of cells were bound by moving potential wells whose basins were sequentially aligned. However, it is noteworthy that cells could skip the potential wells from the back because the 'driving force' decreased again at the far back of the group (*Long, 2019*). This results in leakage of cells in the migratory group (*Holz and Chen, 1978*; *Novickcohen and Segel, 1984*; *Scribner et al., 1974*). Phenotypes with weaker chemotactic abilities were located at the back of the group, where the effective potential well was shallower (*Figure 3C*), allowing for more chances to skip. Thus, such collective migration selects bacteria with higher chemotactic abilities (*Liu et al., 2021*; *Liu et al., 2019*).

The simple computational principle of behavioral modulation to allocate different phenotypes in the collective group is likely not limited to sensing the self-generated signal by consumption of attractant. A prominent example of trail-following migration (*Couzin and Krause, 2003*; *Helbing et al., 1997*) and a typical class of collective behavior is represented by a modified Langevin-type model, where individuals tracing the accumulated signal secreted by all participants (Equation S20) can reproduce similar spatial-temporal dynamics of behavioral modulation, as well as ordered arrangements of phenotypes in the migratory group (*Appendix 1—figure 10*). Thus, this mechanism of matching individual abilities by the signal strength might provide an explanation of how other higher organisms organize ordered structures during group migration.

## Materials and methods

### Strains

The wild-type strain *E. coli* (RP437) and its mutants were used in this study, where all plasmids were kindly provided by Dr Chenli Liu. Specifically, the Tar-titratable strain was constructed by recombineering according to previous research (*Zheng et al., 2016*). Specifically, the DNA cassette of the *Ptet-tetR-tar* feedback loop was amplified and inserted into the chromosomal *attB* site by recombineering with the aid of plasmid *pSim5*. The *tar* gene at the native locus was seamlessly replaced with the *aph* gene by using the same recombineering protocol. To color-code the strains, we use plasmids with chloramphenicol-resistant gene carrying *yfp* under constitutive promoter (for JCY1 strain) and *pLambda*-driven mRFP1 plasmids maintained by kanamycin (for JCY2). To color-code Tar-titratable strain (JCY20), a plasmid carrying *yfp* chloramphenicol-resistant gene was transformed into constructed Tar-titratable strain. The *tar*-only strain (UU1624) was modified from RP734 and was kindly provided by Prof. Johan Sandy Parkinson.

### Media and growth conditions

For bacterial culture, the M9 supplemented medium was used. The preparation of the M9 supplemented medium follows the recipe in previous study (*Fu et al., 2018*): 1×M9 salts, supplemented with

0.4% (v/v) glycerol, 0.1% (w/v) casamino acids, 1.0 mM  magnesium sulfate, and 0.05% (w/v) polyvinylpyrrolidone-40. 1×M9 salts were prepared to be 5×M9 salts stock solution: 33.9 g/L  Na$_2$HPO$_4$, 15 g/L  KH$_2$PO$_4$, 2.5 g/L  NaCl, 5.0 g/L  NH$_4$Cl.

For migration experiments in the micro-channel, the M9 motility buffer was used. The recipe was: 1×M9 salts, supplemented with 0.4% (v/v) glycerol, 1.0 mM magnesium sulfate, and 0.05% (w/v) polyvinylpyrrolidone-40, 0.1 mM  EDTA, 0.01 mM methionine, and supplemented with 200 μM aspartic acid.

For the migration rate measurements, the M9 amino acid medium with 0.2% (w/v) agar was used to prepare swim plate (*Liu et al., 2019*). The recipe was: 1 × M9 salts, supplemented with 0.4% (v/v) glycerol, 1× amino acid, 200 μM aspartic acid, 1.0 mM magnesium sulfate, and 0.05% (w/v) polyvinylpyrrolidone-40. 1× amino acid were prepared to be 5× amino acid stock solution: 4 mM alanine, 26 mM arginine (HCl), 0.5 mM cysteine (HCl·H$_2$O), 3.3 mM glutamic acid (K salt), 3 mM glutamine, 4 mM glycine, 1 mM histidine (HCl·H$_2$O), 2 mM isoleucine, 4 mM leucine,  lysine, 1 mM methionine, 2 mM phenylalanine, 2 mM proline,  threonine, 0.5 mM tryptophane, 1 mM tyrosine, 3 mM valine. All experiments were carried out at 30 °C. Plasmids were maintained by 50 μg/mL kanamycin or 25 μg/mL chloramphenicol.

## Sample preparation

The bacteria from frozen stock were streaked onto the standard Luria-Bertani agar plate with 2 % (w/v) agar and cultured at 37 °C overnight. Three to five separate colonies were picked and inoculated in 2 mL M9 supplemented medium for overnight culture with corresponding antibiotics to maintain plasmids. The overnight culture was diluted by 1:100 into 2 mL M9 supplemented medium the next morning. For Tar titration strains, related aTc were added in this step. When the culture OD600 reaches 0.2–0.25, it was then diluted into pre-warmed 15 mL M9 supplemented medium so that the final OD600 was about 0.05 (*Liu et al., 2019*; *Zheng et al., 2020*; *Zheng et al., 2016*).

Bacteria were washed with the M9 motility buffer and were re-suspended in fresh M9 motility buffer to concentrate cell density at OD600 about 1.0. Then, the wild-type strain and fluorescent strain were mixed with ratio of 400:1 before loaded in the microfluidic chamber (*Fu et al., 2018*; *Saragosti et al., 2011*). For Tar titration experiments, the wild-type strain (RP437) was mixed with two fluorescent strains (JCY2 and JCY20) by 400:8:1.

## Microfabrication

The microfluidic devices were fabricated with the same protocol and the same design as previous research (*Bai et al., 2018*; *Fu et al., 2018*), except that the capillary channel was designed longer than that of previous ones. The size of the main channel was 20 mm×0.6 mm× 0.02 mm and only one gate at the end of the channel was kept (*Appendix 1—figure 1A*).

## Band formation

Fluorescent cells were mixed with non-fluorescent cells by 1:400 for cell tracking in the dense band. Sample of mixed cells with density OD600 ≈1.0 was gently loaded into the microfluidic device and then the device was spun for 15 min at 3000 rpm  in a 30 °C environmental room so that almost 100,000-150,000 cells were placed to the end of the channel. After spinning, the microfluidic device was placed on an inverted microscope (Nikon Ti-E) equipped with a custom environmental chamber set to 50% humidity and 30 °C.

## Imaging

The microscope and its automated stage were controlled by a custom MATLAB script via the μManager interface (*Edelstein et al., 2014*; *Fu et al., 2018*). About 30 min after the sample loading, a 4× objective (Nikon CFI Plan Fluor DL4X F, NA 0.13, WD 16.4 mm , PhL) was placed in the wave front for imaging. The fluorescent bacteria, seen as randomly picked samples of the migrating group, were captured continuously with frame rate of 9 fps in 10 min until they leave the view. Typical tracks are longer than 300 s. Time-lapsed images were acquired by a ZYLA 4.2MP Plus CL10 camera (2048 × 2048 arrays of 6.5 μm×6.5 μm pixels) at 9 frames/s (fps) . An LED illuminator (0034R-X-Cite 110LED) and an EYFP block (Chroma 49003; Ex: ET500/X 20, Em: ET535/30 m) compose the lightening system.

For the Tar titration experiments, the channel was first scanned with 10 × objective (CFI Plan Fluor DL 10 × A, NA 0.30, WD 15.2 mm, PH-1) enlighten by an LED illuminator (0034R-X-Cite 110LED) through the RFP block (Chroma 49005, Ex: ET545/X 30, Em: ET620/60 m) and EYFP block channels for seven neighbored views around the migration group. These images were further combined to two large pictures of the RFP strains and YFP strains. The channel was scanned twice, respectively before and after the 10 min tracking of fluorescent Tar-titrated cells.

### Track extraction and state assignment

The acquired movie was first analyzed with the U-track software package to identify bacteria and to get their trajectories (*Jaqaman et al., 2008*). Then the tracks were labeled by run state and tumble state by a custom MATLAB package (*Waite et al., 2016*) using a previously described clustering algorithm (*Dufour et al., 2016*).

### Track analysis

The group velocity $V_G$ was calculated by averaging the frame-to-frame velocity ($dt \approx 0.11\mathrm{s}$) over all tracks and all time. The cell number for the first frame over a spatial bin of $\Delta x = 60\,\mu\mathrm{m}$ and a channel section $a$=12,000 μm² were calculated to get the density profile $\rho\left(x, t=0\right) = \frac{\sum i(x,t)}{a \cdot dx}$ . The peak position of the first frame ($x_{peak}\left(t=0\right)$) was then determined by the maximum of $\rho\left(x, t=0\right)$. The position of each bacterium ($x_i\left(t\right)$) was transformed to moving coordinate position $z_i$ by the group velocity $V_G$ and origin of the axis on the density peak by $z_i = x_i\left(t\right) - V_G t - x_{peak}\left(t=0\right)$ . Given the relative position of each cell, we recalculated the density profile in moving coordinate $\rho\left(z\right) = \frac{\sum i(z)}{a \cdot dx}$ . The width of the density profile was defined by two times the standard deviation of relative positions $2\sigma = 2\sqrt{\frac{1}{n-1}\sum_{i=1}^{n}\left(z_i - \langle z \rangle\right)^2}$ . The spatial distribution of the instantaneous velocities $\langle V_I\left(z\right)\rangle$ was calculated by averaging the velocity in spatial bin of $\Delta z = 240\,\mu\mathrm{m}$.

A tumble-run event is the minimal element of bacterial behavior. The typical spatial scale of a tumble-run event is about 20 μm, which is much smaller than the spatial bin size chosen in this study (240 μm). The spatial distributions of run time $\langle \tau_R\left(z\right)\rangle$, tumble time $\langle \tau_T\left(z\right)\rangle$, and run length $\langle l_R\left(z\right)\rangle$ were calculated by averaging the related values of all the events with tumbling position ($z_T$) located in each spatial bin ($z$). As the displacement of tumble is small, the tumbling position is approximately the starting position of runs. For each tumble-run event, we have the vector linking starting position and end position of the run. The running angle $\theta_R$ is then defined by the angle between run direction and the group migration direction. One can easily deduce all the other quantities with the formulations in *Table 1*.

**Table 1.** Summary of quantities.

| Quantities | Definition | Formulations |
|---|---|---|
| $V_G$ | Group velocity | $V_G = \left\langle \frac{dx}{dt} \right\rangle$ |
| $z$ | Moving coordinate | $z = x - V_G t$ |
| $V_I\left(z\right)$ | Instantaneous velocity | $V_I\left(z\right) = \left\langle \frac{dx(z)}{dt} \right\rangle$ |
| $V_D\left(z\right)$ | Expected drift velocity | $V_D\left(z\right) = \frac{\langle L_R(z) \cdot \cos\theta_R(z) \rangle}{\langle \tau_R(z) + \tau_T(z) \rangle}$ |
| $\rho\left(z\right)$ | Cell density | $\rho\left(z\right) = \frac{\sum i(z)}{a \cdot \Delta z}$ |
| $S\left(z\right)$ | Chemoattractant concentration | $-V_G \frac{dS}{dz} = D_s \frac{\partial^2 S}{\partial z^2} - k\rho$ |
| $g\left(z\right)$ | Perceived gradient | $g\left(z\right) = \frac{d\ln\left(\frac{1+S(z)/K_{off}}{1+S(z)/K_{on}}\right)}{dz}$ |

## Growth rate and migration rate measurement

Growth rates of Tar-titrated strains were calculated from exponential fitting ($R^2 > 0.99$) over measured curves of cell density (OD600) with respect to time. A 250 mL flask with 20 mL M9 supplement medium were used. All measurements were performed in a vibrator of rotation rate of 150 rpm at 30 °C. OD600 was measured by a spectrophotometer reader every 25 min. Each strain has been measured for at least three times.

The semi-solid agar plate was illuminated from bottom by a circular white LED array with a light box as described previously (*Liu et al., 2011*; *Liu et al., 2019*; *Wolfe and Berg, 1989*) and was imaged at each 2 hr by a camera located on the top. As bacteria swimming in the plate forms 'Adler ring', we used the first maximal cell density from the edge to define the moving edge of bacterial chemotaxis. The migration rate was then calculated from a linear fit over the data of edge positions in respect to time ($R^2 > 0.99$).

## Models and simulations

Details of the theoretical models and numerical simulations were presented in the appendix notes. In which, the Langevin equation was deduced and solved numerically with a particle-based simulation; the approximated OU-type equation and its traveling wave solution were deduced; an agent-based simulation of bacterial with chemotaxis pathway was performed following previous works (*Dufour et al., 2014*; *Jiang et al., 2010*; *Sneddon et al., 2012*).

## Acknowledgements

The authors acknowledge C Liu, for sharing *E. coli* strains and plasmids; S Huang for help with the microfluidics; T Emonet for help with single-cell tracking and agent-based simulations; T Hwa, F Jin, L Luo, J Long for insightful discussions. This work was financially supported by the National Key R&D Program of China (Grant No. 2018YFA0903400), National Natural Science Foundation of China (Grant No. 32071417), CAS Interdisciplinary Innovation Team (Grant No. JCTD-2019-16), Guangdong Provincial Key Laboratory of Synthetic Genomics (Grant No. 2019B030301006), Strategic Priority Research Program of Chinese Academy of Sciences (XDPB1803) to XF, National Natural Science Foundation of China (Grant Nos. 11804355 and 31770111) to YB.

## Additional information

### Funding

| Funder | Grant reference number | Author |
| --- | --- | --- |
| National Key Research and Development Program of China | 2018YFA0903400 | Xiongfei Fu |
| National Natural Science Foundation of China | 32071417 | Xiongfei Fu |
| CAS Interdisciplinary Innovation Team | JCTD-2019-16 | Xiongfei Fu |
| Guangdong Provincial Key Laboratory of Synthetic Genomics | 2019B030301006 | Xiongfei Fu |
| Strategic Priority Research Program of Chinese Academy of Sciences | XDPB1803 | Xiongfei Fu |
| National Natural Science Foundation of China | 11804355 | Yang Bai |
| National Natural Science Foundation of China | 31770111 | Yang Bai |

| Funder | Grant reference number | Author |
| --- | --- | --- |

The funders had no role in study design, data collection and interpretation, or the decision to submit the work for publication.

## Author contributions

Yang Bai, Data curation, Formal analysis, Funding acquisition, Investigation, Writing - original draft; Caiyun He, Data curation, Investigation, Methodology, Writing - original draft; Pan Chu, Methodology; Junjiajia Long, Writing - review and editing; Xuefei Li, Formal analysis, Writing - review and editing; Xiongfei Fu, Conceptualization, Formal analysis, Funding acquisition, Project administration, Resources, Supervision, Writing - review and editing

## Author ORCIDs

Yang Bai http://orcid.org/0000-0001-9976-2686
Xiongfei Fu http://orcid.org/0000-0003-3657-8296

## Decision letter and Author response

Decision letter https://doi.org/10.7554/eLife.67316.sa1
Author response https://doi.org/10.7554/eLife.67316.sa2

# Additional files

## Supplementary files

• Transparent reporting form

## Data availability

All data generated or analysed during this study are included in the manuscript and supporting files.

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

## Appendix 1

### Theory and numerical simulations

Summary

In this note, we first deduced the Langevin-type equation from the classic KS model that describes bacteria group migration in a capillary. The Langevin-type equation describes the dynamics of single-cell particle motion as active particles (Section 1). Together with the chemoattractant consumption equation, we investigated the Langevin-type equation through particle-based simulations for a group of particles of single phenotype or multi-phenotypes (Section 2). Then, we transformed the equations into moving coordinate, so that the stable density profiles as well as the chemoattractant profiles were observed (Section 3). In this moving coordinate, the chemoattractant consumption can be decoupled from particle dynamics, which allows us to simulate particle motions over a non-consumable field of chemoattractant on the move (Section 4). Furthermore, the Langevin dynamics describes that the particle motion was simplified to an OU-type process, of which the particle dynamics can be solved analytically (Section 5). To gain more insights of the chemotactic response, we described an agent-based model which integrated the bacterial chemotaxis pathway in Section 6. And specifically, we deduced the chemotactic ability $\chi$ from the pathway parameters (Section 7). The agent-based simulations were performed under a moving gradient, from which ordered structures of phenotypes were recaptured (Section 8). The model was further extended to system with signal secretion and shows similar results (Section 9).

### 1. The Langevin equation with chemoattractant consumption

The bacterial chemotactic migration in a capillary was first modeled by Keller and Segel in 1971 (**Keller and Segel, 1971a**). In this model, the bacterial cell division was neglected as the nutrition in the capillary was poor. In 1D, the dynamics of cell density $\rho(x, t)$ is a combination of diffusion and chemotaxis:

$$\frac{\partial \rho(x,t)}{\partial t} = D\frac{\partial^2 \rho(x,t)}{\partial x^2} + \chi\frac{\partial}{\partial x}\left(\rho(x,t)\, g(x,t)\right) \tag{S1}$$

where $D$ and $\chi$ respectively represent the diffusion coefficient and chemotactic ability of bacteria, and $g(x, t)$ represents the perceived gradient of chemoattractant.

The dynamics of the chemoattractant $S(x, t)$ follows a reaction-diffusion-type equation with a diffusion coefficient $D_s$ and consumption with a constant rate $k$:

$$\frac{\partial S(x,t)}{\partial t} = D_s\frac{\partial^2 S(x,t)}{\partial x^2} - k\rho(x,t) \tag{S2}$$

From the chemoattractant profile $S(x, t)$, we can get $g(x, t)$ by:

$$g(x,t) = \frac{\partial ln\left(\frac{1+S(x,t)/K_{off}}{1+S(x,t)/K_{on}}\right)}{\partial x} \tag{S3}$$

with $K_{on}$, $K_{off}$ were dissociation constants for active and inactive conformation of the receptor, respectively.

Following **Rosen, 1973**, one can deduce the Fokker-Planck equation associated to the KS model that describes the probability $P = P(x, y; t)$ to observe of a cell which initiates from position $y$ at position $x$ and time $t$. The related equation writes:

$$\frac{\partial P}{\partial t} = D\frac{\partial^2 P}{\partial x^2} + \chi\frac{\partial(Pg(x,t))}{\partial x} \tag{S4}$$

Equation S4 is equivalent to the Langevin dynamics as:

$$dx_i = \chi g(x_i, t)\, dt + \epsilon dW \tag{S5}$$

where $x_i$ represents the position of the $i$-th bacteria and $g(x_i, t)$ is the perceived gradient of each bacteria at time $t$. $dW$ is a Wiener process that describes a random walk with noise level $\epsilon = \sqrt{2D}$.

As previously proofed, the expected drift velocity equals the production of chemotactic ability $\chi$ and the perceived gradient $g$, $V_D = \chi g$ (**Waite et al., 2016**). So that, the Langevin-type Equation S5 can be reformed to:

$$dx_i = V_{D,i}dt + \epsilon dW \tag{S6}$$

The initial condition of the chemoattractant was $S(x,0) = S_0 = 200 \ \mu$M. In the stochastic model, the density profile $\rho(x,t)$ was calculated by the sum of cell number ($\sum i(x,t)$) in certain location $[x, x + \Delta x]$ and time $t$ over volume $a \cdot \Delta x$, where $a$ is the capillary section area and $dx$ is the spatial bin size:

$$\rho(x,t) = \frac{\sum i(x,t)}{a \cdot \Delta x} \tag{S7}$$

## 2. Particle-based simulation with consumption

The Langevin-type model in the absolute coordinate $x$ (Equation S5) was simulated with a customized code using MATLAB. In each case, 100,000 particles were simulated in a 1D space where $x \in [0, 20]$ mm. The absorption boundaries were applied for $x = 0$ mm and $x = 20$ mm, so that particles stay at the boundary until they leave them actively ($x(x > 0 \text{ mm}) = 0$ and $x(x > 20 \text{ mm}) = 20$ mm). Initially all particles were placed at $x = 0$ mm and their positions were calculated according to Equation S7 at each time step of $\Delta t = 0.05s$ for 60 min. The random noise follows a Gaussian distribution generated by MATLAB function 'randn.m' with amplitude multiplied by $\sqrt{2}\epsilon\sqrt{\Delta t}$ .

Unlike the particles that change their position continuously, the spatial profile of cell density $\rho(x,t)$ was first calculated on a space mesh of grid $\Delta x = 10 \ \mu$m, and then linearly interpolated to a finer mesh of grid $\Delta x = 0.1 \ \mu$m. The chemoattractant $S(x,t)$ and perceived gradient $g(x,t)$ were deduced accordingly. At each time point, following Equation S7, cells located on grid $[x, x + \Delta x]$ were summed and were divided over $a\Delta x$ to get the density profile. In order to avoid fast consumption near $S = 0$, a saturation effect of consumption rate was added to Equation S3 using $S/(S + K_s)$.

$$\frac{\partial S(x,t)}{\partial t} = D_s \frac{\partial^2 S(x,t)}{\partial x^2} - k\rho(x,t) \frac{S(x,t)}{S(x,t) + K_s} \tag{S8}$$

where $K_s = 0.1 \ \mu$M, of which the value was small and does not affect the linear relation in most range of $S > K_s$ .

Then the chemoattractant profile $S(x,t)$ was calculated from $\rho(x,t)$ according to Equation S8, using a fourth-order Runge-Kutta method from an initial condition of $S(x,0) = S_0$ . The perceived gradient $g(x,t)$ was further calculated from $S(x,t)$ on Equation S3. Having the discrete $g(x,t)$ over grid of $\Delta x = 0.1 \ \mu$m, the perceived gradient of each particle $g(x_i)$ was assigned by detecting the particle position over the spatial mesh grid.

Following the above instructions, density profile and peak positions with bacteria of $\chi = 0.11 \text{ mm}^2\text{min}^{-1}$ and $D = \chi/22$ were plotted in *Appendix 1—figure 5A*. The group migration reached steady state after 20 min simulation. The peak velocity was then defined by linear fitting of peak positions for $t \in [20, 60]$ min, which was found to be $V_p = 3.03 \ \mu$m/s.

To simulate a bacterial group of multiple phenotypes, we constructed 100,000 cells with their chemotactic ability $\chi$ follows a Gaussian distribution with the average chemotactic ability $\langle \chi \rangle = 0.11 \text{ mm}^2\text{min}^{-1}$ and s.d. of $0.03 \text{ mm}^2\text{min}^{-1}$ (*Appendix 1—figure 5B*) while the diffusion coefficient $D$ was fixed to $\langle \chi \rangle /22$ for all cells. The consumption rate of each bacterium was also set constant of $4.6 \cdot 10^{-9} \ \mu$M/cell/s. The group of multi-phenotypes migrates slower than single phenotype with peak velocity $V_p = 2.90 \ \mu$m/s (*Appendix 1—figure 5A*). The bacterial positions after 20 min simulation were further transformed to the moving coordinate $z_i = x_i - V_p t$. They were subgrouped according to their chemotactic ability $\chi$ to six subgroups. And the corresponding average density profiles of each subgroups were plotted in *Appendix 1—figure 5D* with colors defined in *Appendix 1—figure 5C*. The density profile of subgroups was found ordered according to their average chemotactic ability $\chi$. The time-shifted profiles, $S(z)$ and $g(z)$, were shown in *Appendix 1—figure 5E,F*.

The simulations of bacteria with consumption confirmed that coherent group migration and ordered spatial structures can emerge from a system of active particles modulated by gradient of consumable chemoattractant.

## 3. Traveling wave solution

As the density profile was found stable during the experimental time scale (30 min), we are allowed to use a moving coordinate $z = x - V_G t$ in searching for a traveling wave solution. Equation S2, Equation S5, Equation S7 become:

$$dz_i + V_G dt = \chi g\left(z_i\right) dt + \epsilon dW \tag{S9}$$

$$-V_G \frac{dS(z)}{dz} = D_s \frac{d^2 S(z)}{dz^2} - k\rho\left(z\right) \tag{S10}$$

$$\rho\left(z\right) = \frac{\sum i(z)}{a \cdot \Delta z} \tag{S11}$$

Given a stable density profile measured experimentally, we can integrate Equation S10 to get a stable profile of chemoattractant $S\left(z\right)$. As the chemoattractant was unconsumed at $z = \infty$ and it is completely consumed as the wave passes $z = -\infty$, the boundary condition of $S\left(z\right)$ can be written as $S\left(\infty\right) = S_0$ and $S\left(-\infty\right) = 0$. We get:

$$S\left(z\right) = S_0 - \frac{k}{D_s} e^{-\frac{V_G}{D_s}z} \int_z^\infty M\left(z\right) e^{\frac{V_G}{D_s}z} dz \tag{S12}$$

where $M\left(z\right) = \int_z^\infty \rho\left(z\right) dz$ is the accumulated number of cells up to position $z$. We then used the experimentally measured cell density profile together with the group speed $V_G$ to calculate the moving attractant profile $S\left(z\right)$ (**Appendix 1—figure 5A**). The perceived gradient $g\left(z\right)$ in the moving coordinate was then calculated by:

$$g\left(z\right) = dln \left( \frac{1 + S(z)/K_{off}}{1 + S(z)/K_{on}} \right) /dz \tag{S13}$$

In a stable migration group, the $g\left(z\right)$ deceases from back to front over the density profile. Suppose the gradient does not steepen toward the back, then cells with lower chemotactic abilities will fall behind and see shallower gradient, drifting even slower and causing the wave to spread out. Since the gradient is formed by local consumption, the front of the wave will have less cells and become shallower, while the back of the wave will accumulate more cells and become steeper. This will automatically readjust the gradient steepness until the back becomes steep enough to allow cells will lower chemotactic abilities to catch up, stabilizing the gradient and the traveling wave.

## 4. Particle-based model with preset moving gradient

The chemoattractant profile $S\left(z\right)$ can be deduced by an experimentally measured stable density profile $\rho\left(z\right)$ in the moving coordinate. With the experimental mesh grid $\Delta x = 240 \, \mu$m, we plot the average $S\left(z\right)$ over three biological replicates in **Appendix 1—figure 4D** (circles). In order to apply this profile into the particle-based simulation, we first expanded the space range of $S\left(z\right)$ to $x \in \left[-4800, 4800\right] \, \mu$m by setting $S\left(z < -2400 \, \mu\text{m}\right) = 0$ and $S\left(z > 2400 \, \mu\text{m}\right) = S_0$. Next, we used a cubic spline data interpolation to construct a profile of $S\left(z\right)$ with much finer spatial mesh grid $\Delta x = 0.001 \, \mu$m (**Appendix 1—figure 4D**, line). Therefore, a $g\left(z\right)$ profile on finer meshes was calculated accordingly (**Figure 2A**). The $g\left(z\right)$ profile was then translated to an absolute coordinate by $g\left(x, t\right) = g\left(z\right) + V_G t$, with $V_G = 3.0 \, \mu$m/s.

Using the preset moving gradient $g\left(x, t\right)$, we can simulate the particle motions without calculation of $\rho\left(x, t\right)$ and $S\left(x, t\right)$. We used the same method as described in Section 2 to simulate particle motions, and at each time point $t$, we assigned $g\left(x_i\right)$ by indexing bacterial position on $g\left(x, t\right)$, with spatial precision of $\Delta x = 0.001 \, \mu$m.

For bacteria of four different $\chi$ ranging from 0.15 to 0.3 mm²min⁻¹, we simulated 10,000 particles for each phenotype with the same noise level $\epsilon = 15 \, \text{mm} \cdot \text{min}^{-0.5}$. During simulation of 60 min, bacteria of all phenotypes move as a group with peak positions linearly increasing with time. The fitted slope of all peaks equals the preset velocity of the moving gradient $V_G$. In the moving coordinate $z$, the density profiles of all phenotypes were ordered (**Figure 2A**), and they were

clearly ordered by $\chi$. The peak positions and mean positions of the density profile increase with $\chi$ (**Figure 2C**), while the width (defined by $\sigma$) decreases with $\chi$ (**Figure 2D**).

## 5. The type model

The perceived gradient $g(z)$ deduced from experimental data shows linear decreasing trend in the major range of density profile $-0.4$ mm $< z < 0.4$ mm. It can be then linearly fitted by $g(z) \approx g_1 z + g_0$ with $g_1 \approx -2.1$ mm$^{-2}$ and $g_0 \approx 1.2$ mm$^{-1}$. Subscribe this linear approximation in Equation S7, we get:

$$dz_i = \chi g_1 z_i dt + (\chi g_0 - V_G) dt + \epsilon dW \tag{S14}$$

Now, we compare Equation S10 to the standard OU model given by $dx = -\lambda x dt + \epsilon dW$. Equation S9 can be taken as an OU-type process with $-\chi g_1$ as the reversion rate $\lambda$ and the mean position deviated from origin to $z_0$:

$$z_0 = \frac{\chi g_0 - V_G}{\chi g_1} \tag{S15}$$

where the mean position $z_0$ is the steady-state position of its ensemble average $\langle z \rangle$:

$$d \langle z \rangle = \chi g_1 \langle z \rangle dt + (\chi g_0 - V_G) dt \tag{S16}$$

The original OU model describes an active particle motion under a linear force field around $x = 0$. Since Equation S10 can be taken as a generalized OU model, we can thus consider our bacteria in the migration group as an active particle under an effective force field of $\chi g_1 z + (\chi g_0 - V_G)$. Integrating this force field over $z$, we get an effective potential well $U(z)$:

$$U(z) = \int (\chi |g_1| z + (\chi g_0 - V_G)) dz = \frac{1}{2} \chi |g_1| z^2 + (\chi g_0 - V_G) z + U_0 \tag{S17}$$

The $U_0$ is an integral constant controlled by boundary condition, which was set by $U(\infty) = 0$ in this article. The effective potential well $U(z)$ generated by $U(z) = \int (\chi g(z) - V_G) dz$ from the particle-based simulation and Equation S17 are compared in **Figure 2B**.

Using a variable substitution of $\widetilde{x} = z - (\chi g_0 - V_G)/g_1$, and applying the general solution of the standard OU model of $\widetilde{x}(t) = \widetilde{x}_0 e^{-\lambda t} + \int_0^t \epsilon dW \cdot e^{-\lambda(t-t')} dt'$ (**Cherstvy et al., 2018**), we get the mean square displacement: $MSD \equiv \left\langle (z(t) - z_0)^2 \right\rangle = \langle z_0 \rangle^2 (1 - e^{-\chi g_1 t})^2 + \frac{\epsilon^2}{2\chi g_1} (1 - e^{-2\chi g_1 t})$. The standard distribution $\sigma$ of the density distribution is the limit of MSD:

$$\sigma = \langle z_0 \rangle^2 + \frac{\epsilon^2}{2\chi g_1} \tag{S18}$$

The trend of both mean positions (**Figure 2C**) and width (**Figure 2D**) was successfully predicted by analytical solutions of the OU-type model Equation S15 and Equation S18.

Moreover, the OU-type model Equation S14 has an equivalent Fokker-Planck equation that describes the probability density function of particles on $z$, $P(z,t)$:

$$\frac{\partial P(z,t)}{\partial t} = \frac{\partial}{\partial z} ((\chi g_1 z + (\chi g_0 - V_G)) P(z,t)) + D \frac{\partial^2 P(z,t)}{\partial z^2} \tag{S19}$$

At $t \to \infty$, the probability density function reaches the steady stat $P(z)$, where the density profile follows $0 = \frac{d}{dz} ((\chi g_1 z_i + (\chi g_0 - V_G)) P(z)) + D \frac{d^2 P(z)}{dz^2}$. Solving this equation with boundary conditions of $P(z = \pm\infty) = 0$ and $\frac{\partial P(z = \pm\infty)}{\partial z} = 0$, we get the steady stat density profile $P(z) = e^{-\frac{1}{D}(\frac{1}{2}\chi g_1 z^2 + (\chi g_0 - V_G) z + C_0)}$ with $C_0$ is the integral constant that allows $\int_{-\infty}^{\infty} P(z) dz = 1$. The exponent of this density profile is position-dependent that drops in parabolic form as $z \to \infty$. Compare to the front of Fisher wave, where the exponent of the density profile decreases linearly with $z$ ($P(z) = e^{-Cz}$), we know that the density profile of a chemotaxis migration wave is more compact than a Fisher wave.

## 6. Agent-based simulation with chemotaxis pathway

In order to simulate more details of bacterial group migration in a capillary, we use a previously established chemotactic pathway integrated agent-based simulation. This model integrated bacterial chemotactic signaling pathway to determine the switch rate of motors, the conformation changes of flagella, as well as the competition of multiple flagella.

We first simulated 80,000 cells evenly distributed in four phenotypes of different receptor gain $N$ with consumption of chemoattractant. The cells were simulated as described in a previous paper (*Saragosti et al., 2011*), while the chemoattractant field $S$ was calculated at each time step ($\Delta t = 0.01$ s) according to Equation S8. In a 3D channel of 0.2 mm×0.01 mm× 10 mm, space was discretized to grid of 2.5 μm×2.5 μm× 2.5 μm . The evolution of the attractant during each time step was calculated by the sum of the consumption of every cell in the neighbor grids weighted by their effective diffusion distance. The consumption rate was fitted for reasonable migration speed. Detailed methods were described in a previous paper (*Xue and Othmer, 2009*). The perceived attractant of each cell was then linearly interpolated from the closest grids of the attractant field.

Simulation starts by setting all cells in one end of the channel ($x = 0$), randomized in $y$ and $z$ directions and setting uniform chemoattractant concentration $S_0 = 200\,\mu$M within the channel. Flip boundary condition was used for all three directions. After 20 min simulation, the bacterial population from stable migration groups (*Appendix 1—figure 6A*) tracks with cellular run-tumble stat was recorded from 30 to 32 min with time step of 0.1 s. Using the group velocity $V_G$ of this period as the speed of moving coordinate $z = x - V_G t$, density distribution of subgroups of different receptor gain $N$ was plotted to show spatial ordering of subgroups (*Appendix 1—figure 6B*). The tracks were analyzed with exactly the same method as the experimental tracks (see Materials and methods). The expected drift velocity $V_D(z)$ of all subgroups decreases from back to front and was also spatially ordered (*Appendix 1—figure 6C*).

The particle-based simulations (Section 4) have already shown that the chemoattractant consumption can be decoupled from bacterial motions. To simplify the analysis, we used the same non-consumable moving gradient $g(x, t)$ described in *Figure 2A*. Using the preset gradient, we simulated 100,000 bacteria of four evenly distribution phenotypes of different receptor gain $N$ or adaptation time $\tau$ or basal CheY-p level $Y_{p0}$ . The simulation was performed in 3D with reflection boundary on $x = 0$ mm & x = 20 mm and with free boundaries in the other two directions ($y$, $z$). Initially all cells were placed on $x = 2$ mm, y = z = 0 mm. In each time point with step $\Delta t = 0.01s$, the internal states of every bacterium were updated according to their perceived gradient indexed by $g(x_i)$ with spatial precision $\Delta x = 0.001\,\mu$m. And then the bacterial moving stat (run or tumble) was updated accordingly, which results in movement to the next time point. Most parameters used in the simulation follow previous paper (*Saragosti et al., 2011*), while the changed parameters were the upper limit of chemoattractant sensing $K_{on} = 1000\,\mu$M (*Fu et al., 2018*); the lower limit of chemoattractant sensing $K_{off} = 1\,\mu$M (*Si et al., 2012*); the diffusion coefficient of chemoattractant $D_s = 1000\,\mu$m$^2$/s (*Fu et al., 2018*); the run speed of bacteria $v_0 = 30\,\mu$m/s (*Keller and Segel, 1971b*); the adaptation time $\tau_M = 1s$ (*Waite et al., 2016*); the motor number was set 5 as in *Sneddon et al., 2012*; while the basic CheY-P level $Yp_0 = 2.77\,\mu$M, which corresponds to $TB_0 \approx 0.27$. After 40 min of simulation, bacterial tracks were recorded with 10 fps for 20 min.

## 7. Estimation of $\chi$ from chemotaxis pathway

In order to compare the agent-based simulation results to the analytical OU-type model, we need to estimate the chemotactic ability $\chi$ from the parameters of chemotaxis pathway. Considering an simple case where perceived gradient of chemoattractant $g$ is a fixed value, the averaged drift velocity of a group of bacteria $V_D$ can be estimated by the parameters of bacterial chemotaxis pathway (*Dufour et al., 2014*): $V_D \approx \frac{\tau'_{R0}}{1+\tau_{R0}/\tau_M} \frac{(1-TB_0)v_0^2 Ng}{d}$ . In this equation, $\tau_{R0}$ is the run time in homogenous environment, $\tau'_{R0} = d\tau_{R0}/df$ with $f$ is the free energy of the receptor, $\tau_M$ is the adaptation time, $d$ is the dimension (e.g. d is 3 in this case), $TB_0$ is the tumble bias in homogenous environment, and $N$ is the receptor gain.

On the other hand, given the expression of $V_D$ , comparing the two expressions of expected drift velocity in fixed gradient, we can deduce:

$$\chi \approx \frac{\tau'_{R0}}{1+\tau_{R0}/\tau_M} \frac{(1-TB_0)v_0^2 N}{d} \tag{S20}$$

Using the expression Equation S20, we can deduce the required receptor gain $N$ from $\chi$, which is the case shown in the main text in *Figure 3*.

## 8. Agent-based simulation for different parameters

To investigate the bacterial behavior in moving gradient, we have simulated bacteria with $\chi$ ranging from 0.15 to 0.30 corresponding to receptor gain $N$ ranging from 14.5 to 25.4. Related results were shown in *Figure 3* and discussed in the main text.

Receptor gain is not the only parameter that affects the chemotactic ability. We have also varied the adaptation time $\tau_M$ and basal CheY-p level $Y_{p0}$ to investigate the effect of these parameters on bacterial behaviors with a moving chemoattractant profile of larger slope of perceived gradient, $g_1 = -2.5 \text{ mm}^{-2}$. For both cases, the density profile $\rho(z)$ spatially ordered according to the variant parameter, and the expected velocities for call parameters $V_d(z)$ decrease from back to front. These results show that bacteria perform mean reversion flow for all these parameters and were spatially ordered according to phenotypes.

For the case of basic *CheY-p* level $Y_{p0}$ ranging from 2.70 μM to 2.85 μM, the basic tumble bias $TB_0$ varies from 0.07–0.19, and the chemotactic ability $\chi$ decreases with $Y_{p0}$ from 0.16 to 0.25 mm²min⁻¹. The mean positions of subgroups increase with $\chi$ so that it decreases with $Y_{p0}$. Unlike the receptor gain $N$, the basic tumble bias also increases the diffusion coefficient $D$. Analogized to the OU-type model, varying $Y_{p0}$ changes the reversion rate $\chi g$ and the noise level $\epsilon = \sqrt{2D}$ at the same time. Thus, the reversion rate decreases with $Y_{p0}$ and the width of the density profile $\sigma$ increases with the reversion rate $r$ (*Appendix 1—figure 8A–D*).

For the case of adaptation time $\tau_M$ ranging from 1 to 10 s, mean positions and width of the density profile show non-monotonic dependence on $\tau_M$. This phenomenon shows that adaptation time has a trade-off effect on bacterial chemotaxis. As *Frankel et al., 2014* have discussed, the increasing adaptation time enforces chemotactic ability, while it weakens the sensitivity of bacteria to the chemoattractant gradient changes. Such dual effect was also reflected on the reversion rate defined by the slope of $V_D$ (*Appendix 1—figure 8E–H*).

## 9. Modified Langevin-type model for accumulated signals

As discussed in the main text, the reversion flow of individuals in the migrating group relies on a decreasing driving force from back to front. In the case of bacterial chemotaxis, such decreasing force comes from the gradient of perceived signal of chemoattractant, which is consumed by all the agents. Such mechanism can be generalized to the systems of trail-following migration (e.g. ants), where the signal is secreted by agents instead of being consumed. In this case, the signal concentration $S_s(x, t)$ decreases from back to front, because more agents have passed the back than front, thus have secreted more signal on the back. Such dynamics can be modeled by similar Langevin equations and signal accumulations as discussed in Section 1:

$$dx_i = \chi_i S dt + \epsilon dW$$

$$\frac{\partial S_s}{\partial t} = \frac{D_s \partial^2 S_s}{\partial x^2} + \gamma \rho - \beta S_s$$

$$\rho(x, t) = \frac{\sum i(x,t)}{a \cdot \Delta x} \tag{S21}$$

where $\gamma = 3 * 10^{-13} \ \mu\text{M} \cdot \text{s}^{-1} \cdot \text{cell}^{-1}$ is the secretion rate of the signal $S_s$ and $\beta = 0.001 \ s^{-1}$ is the decay rate of it. Other terms and parameters are identical to equations in Section 1.

Then, we solve Equation S21 numerically by applying $10^6$ particles with normally distributed response ability $\chi_i$ of mean value of 1 and s.d. of 0.3 (*Appendix 1—figure 10C*). Results show that a coherent migrating group is formed spontaneously with constant migration speed ~6.7 μm/s (*Appendix 1—figure 10B*). In the moving coordinate, the 'effective global force' $S_s(z)$ decreases from the back to front as predicted. The particles with different response ability $\chi_i$ are ordered arranged (*Appendix 1—figure 10D*), while particles with larger $\chi$ locate more in the front.

**Appendix 1—table 1.** Strains and plasmids used in this study.

| Strains | Description | Source |
|---|---|---|
| RP437 | Wild-type strain for chemotaxis | *Fu et al., 2018* |
| JCY1 | RP437+ a Cm$^r$ plasmid with YFP | This study |
| JCY2 | RP437+ a Kan$^r$ plasmid with mRFP1 | This study |
| JCY20 | Construction based on RP437, *tar<> loxp, bla:P*$_{tet}$*-tetR-tar* at *attB* site,+ a Cm$^r$ plasmid with YFP | This study |
| UU1624 | Δaer-1 Δtsr-7028 Δtap-3654 Δtrg-100 | *Gosink et al., 2006* |

**Appendix 1—table 2.** Summary of parameters in agent-based simulation.

| Symbol | Definition | Value | Source |
|---|---|---|---|
| $\chi$ | Chemotaxis ability | Defined in context | This study |
| $k$ | Consumption rate of chemoattractant | Fitted parameter for migration speed | This study |
| $K_{on}$ | Upper limit of chemoattractant sensing | 1 mM | *Fu et al., 2018* |
| $K_{off}$ | Lower limit of chemoattractant sensing | 1 µM | *Si et al., 2012* |
| $D_s$ | Diffusion coefficient of chemoattractant | 1000 µm$^2$/s | *Fu et al., 2018* |
| $v_0$ | Run speed of bacteria | 30 µm/s | *Keller and Segel, 1971b* |
| $\tau_M$ | Adaptation time | 1 s | *Waite et al., 2016* |
| $N$ | Receptor gain | Deduced from chemotactic ability | SI notes in Section 7 |
| $M$ | Motor number | 5 | *Sneddon et al., 2012* |
| $Yp_0$ | Basic CheY-P level | 2.77 µM | Correspond to $TB_0 \approx 0.27$ |

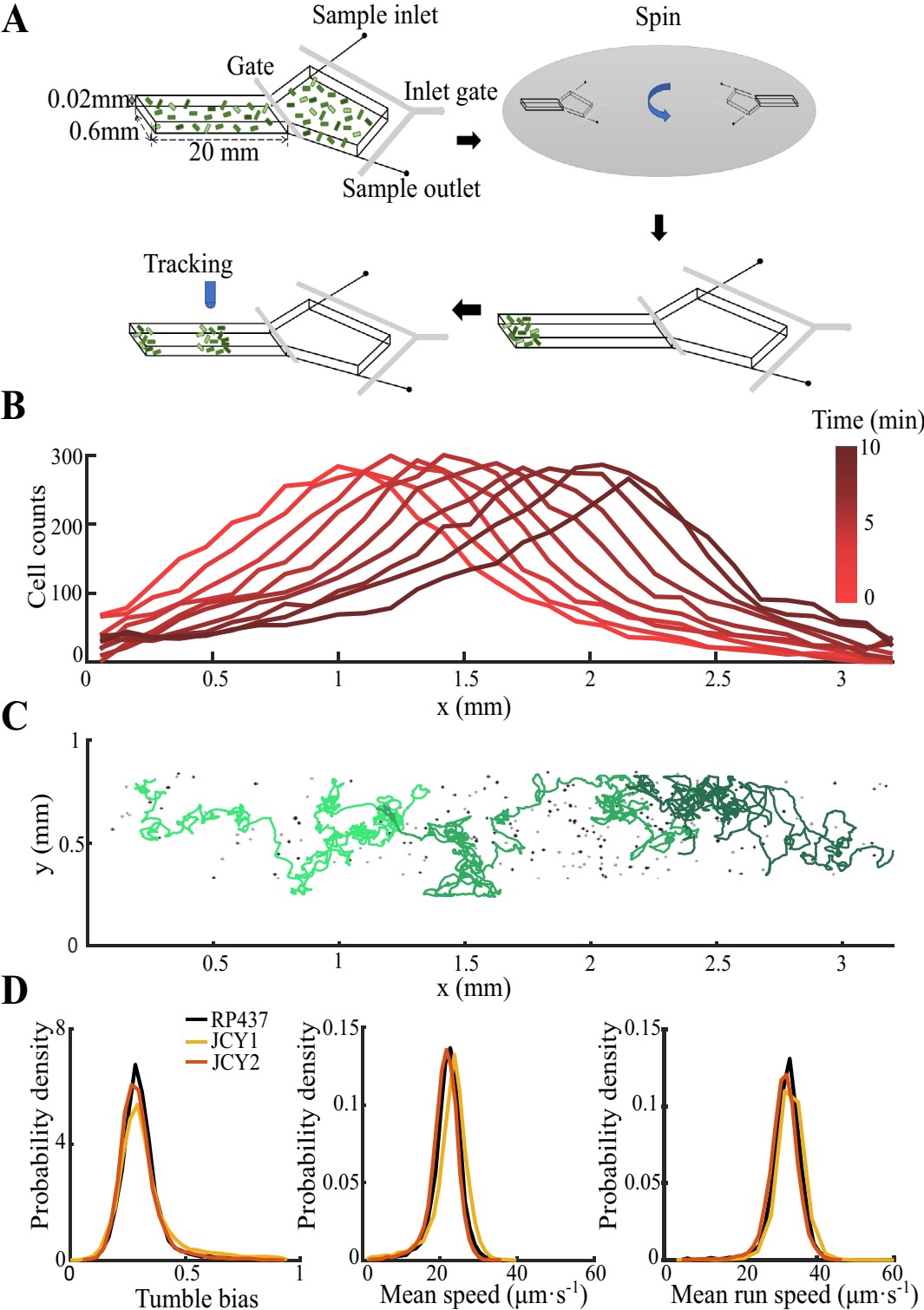

**Appendix 1—figure 1.** Experiment protocol, sample tracks, and fluorescent labeled cells. (**A**) The microfluidic chip used in this study is consisted of a long channel (20 mm×0.6 mm×0.02 mm) and a large chamber (3 mm× 3 mm). Prepared bacteria are first loaded in the chip through inlets with M9 motility buffer (see Materials and methods). Next, the chip is spun at 3000 rpm for 15 min so that the bacteria are concentrated to one end of the channel. Bacteria consume aspartate (Asp) as the single chemoattractant to form a moving gradient that attracts the group migration. Finally, the bacterial

*Appendix 1—figure 1 continued on next page*

*Appendix 1—figure 1 continued*

motions are tracked under microscope of 4× objective by frame rate of 9 fps. (**B**) Cells expressing fluorescent protein (strain JCY2) are premixed into strain RP437 by 1:400 and counted each frame. Cell counts in nine successive frames (1s) were plotted within bin size of $\Delta x = 100\,\mu$m. Colors from light to dark red represent time from 1 to 10 min. (**C**) Sample tracks of bacteria (colored lines) and bacterial signal (black dots) on the last captured frame. (**D**) Fluorescent labeled cells (strain JCY1 and JCY2) and the wild-type strain (RP437) were tracked separately on growth medium. All three strains show the same distribution of tumble bias (left), mean speed (middle), and mean run speed (right) of each track. All these quantities are average of more than 9000 tracks, weighted by their trajectory time. Each curve show data of one experiment.

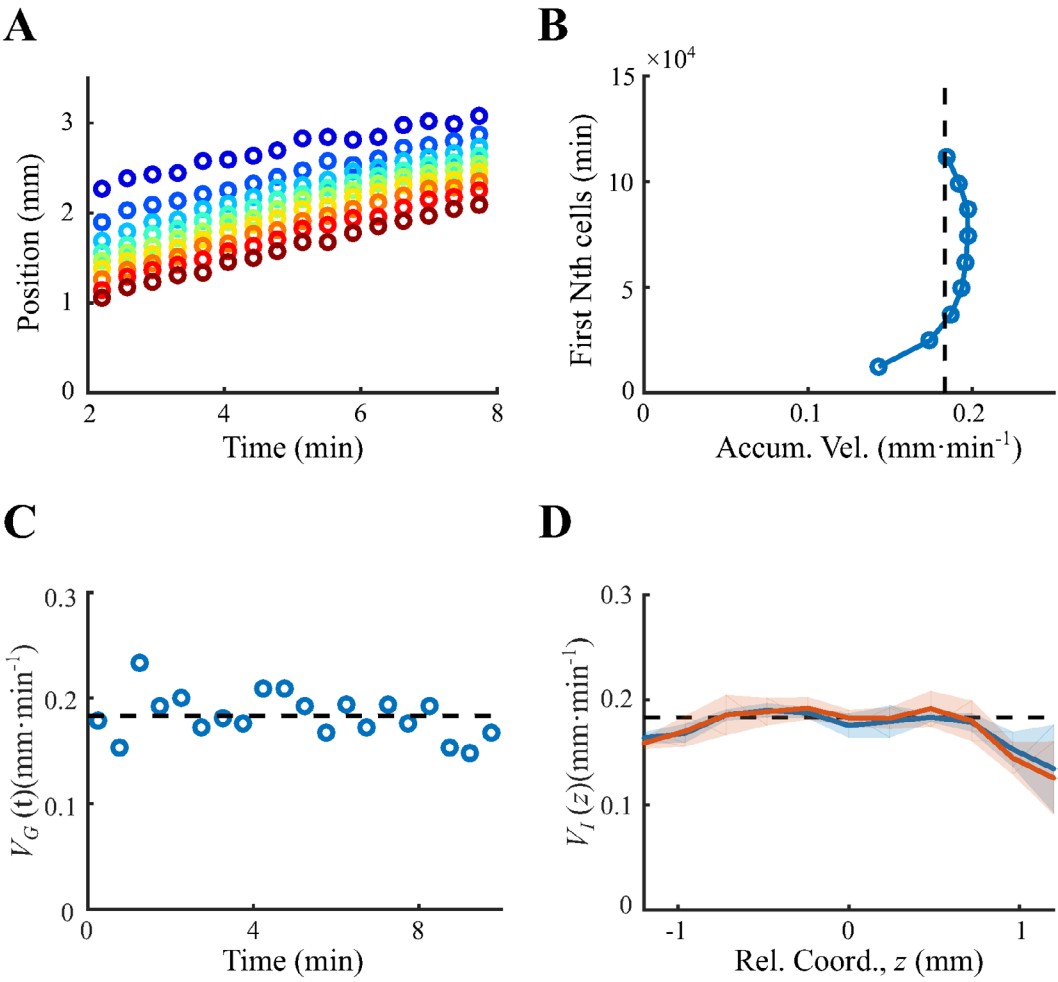

**Appendix 1—figure 2.** Determination of coherent migration group. (**A**) Positions of first $n$-th cell (counted from the front, from blue to red $n = \begin{bmatrix} 1.24\ 2.48\ 3.72\ 4.96\ 6.2\ 7.44\ 8.68\ 9.92\ 1.16 \end{bmatrix} \times 10^4$) increase linearly with time, of which the slope gives the traveling speeds. (**B**) The traveling speeds of the first $n$-th cells are close to the group velocity $V_G$ (black dashed line), representing a coherent migration group. (**C**). The instantaneous group velocity $V_G(t)$ was constant over time. $V_G(t)$ was averaged for all tracks over 30 s (blue dots). Black dashed line represents the average group velocity for all tracks on all times $V_G$ . (**D**) The constant instantaneous velocity $V_I(z) = \langle \Delta x(z)/\Delta t \rangle$ for sampling time $\Delta t = 1\,$s (blue solid line) and $\Delta t = 10\,$s (red solid line), where shaded area was s.e.m. of three replicates. Both curves equal to the group velocity $V_G$ , indicating the migration group is coherent. Data in (**A–C**) was extracted from one experiment.

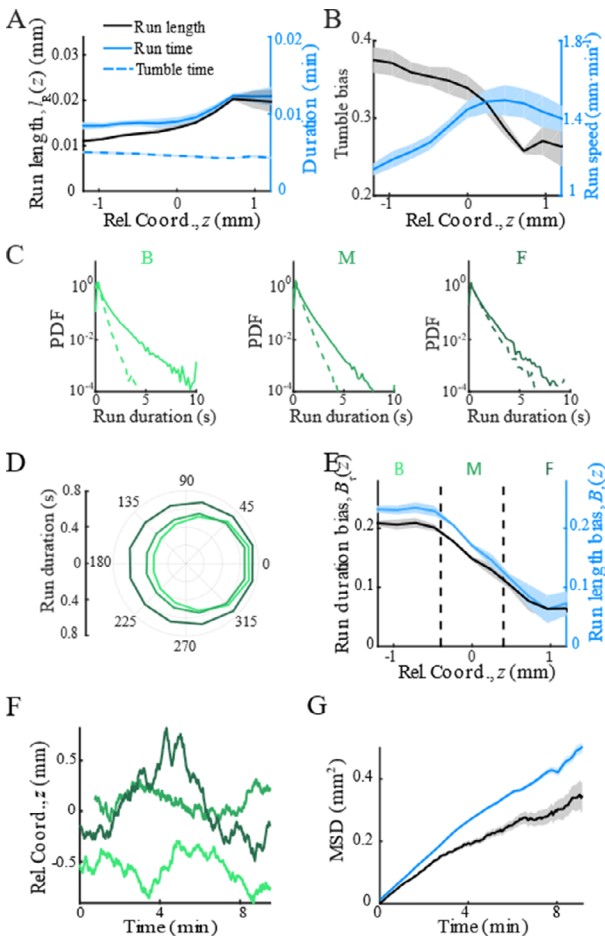

**Appendix 1—figure 3.** Modulation of bacterial behaviors. (**A**) Distribution of forward run duration (solid line) and backward run duration (dash line) in three regions defined in *Figure 1D* and *Appendix 1—figure 3E*. (**B**) Mean run duration in different directions of bacteria in three regions. s.e.m. of four replicates is smaller than thickness of lines. Angular bin size is 15°. (**C**) Spatial structure of adapted tumbled bias (blue) and run speed (red). (**D**) The mean run length in different directions, with the angular bin size of 15°, also shows that cells in the back were better skewed to run forward. (**E**) The run length bias $B_l\left(z\right) = \frac{\langle l_R(z) \cdot \cos\theta_R(z)\rangle}{\langle l_R(z)\rangle}$ (black solid line) and the run time bias $B_\tau\left(z\right) = \frac{\langle \tau_R(z) \cdot \cos\theta_R(z)\rangle}{\langle \tau_R(z)\rangle}$ (blue solid line) both decreased from the back to the front of the migration group. In panels C–E, shaded area represents s.e.m. of three biological replicates. The spatial bin size is 240 µm. (**F**) Representative examples of single-cell trajectories (three colors represent three different tracks) showed the reversion behavior of bacteria around their mean positions. (**G**) The mean reversion behavior results in limited diffusion of bacteria in the moving frame. So that the mean square displacement (MSD) in the direction of migration (black curve) was lower than the pure diffusion of bacteria measured on free liquid with M9 motility buffer (blue curve). The shaded area on black curve represents the s.e.m. of more than 2900 tracks, while the shaded area on blue curve represents the s.e.m. of more than 1900 tracks.

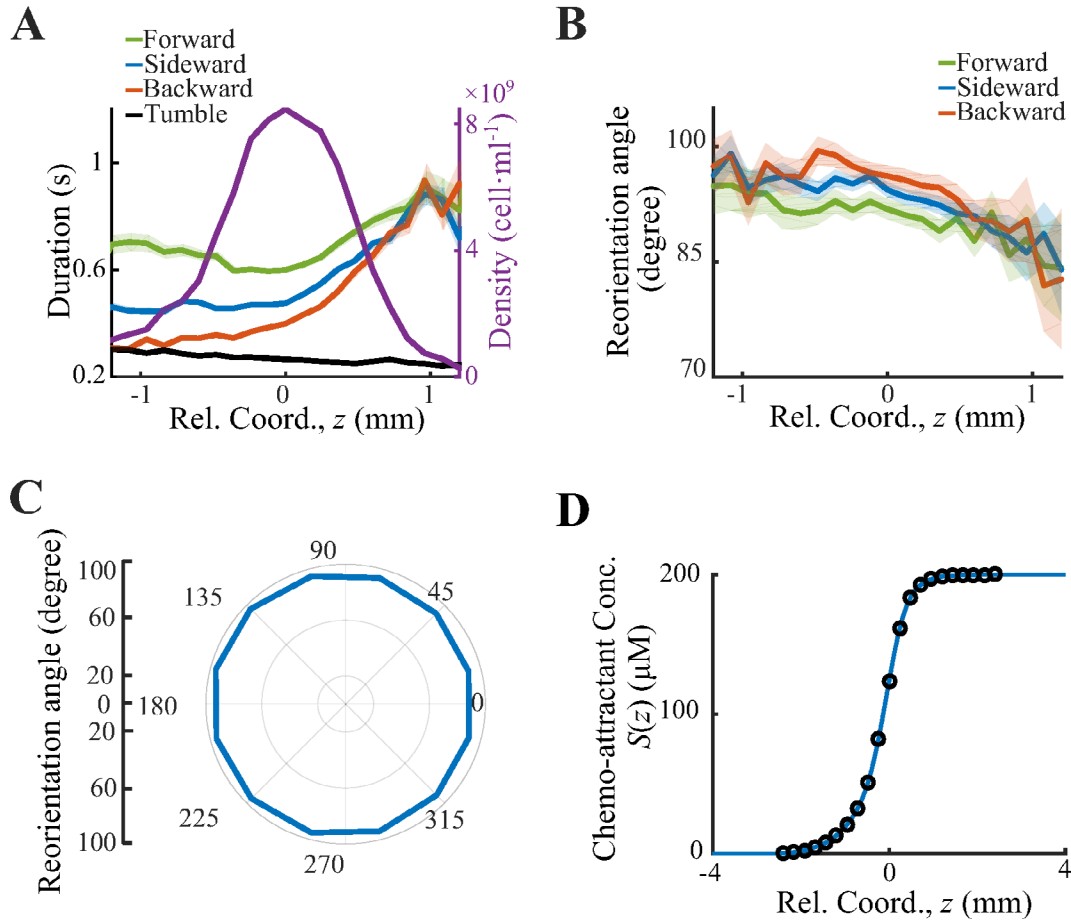

**Appendix 1—figure 4.** Behavior analysis and deduction of the chemoattractant profile. (**A**) All cell tracks are divided into three sectors: forward ($|\theta| \leq 45°$) (green line), sideward ($45° < |\theta| < 135°$) (blue line), backward ($|\theta| \geq 135°$) (red line). The run durations in all three sectors increase form back to front. (**B**) Spatial distribution of reorientation angle. (**C**) Reorientation angles defined in *Saragosti et al., 2011* are plotted over the direction of previous runs. Such angles are biased to the back, indicating that bacteria running down the gradient reorient more in the next tumble. Shaded area shows s.e.m. of 14,892 tracks. The s.e.m. in panel C is smaller than the thickness of the curve. (**D**) The chemoattractant profile $S(z)$ used for the simulation is interpolated by $\Delta z = 0.001\,\text{mm}$ (blue line) from the experimental $S(z)$ (black circles). While the experimental chemoattractant is deduced from the averaged density profile $\rho(z)$ (*Figure 1B*).

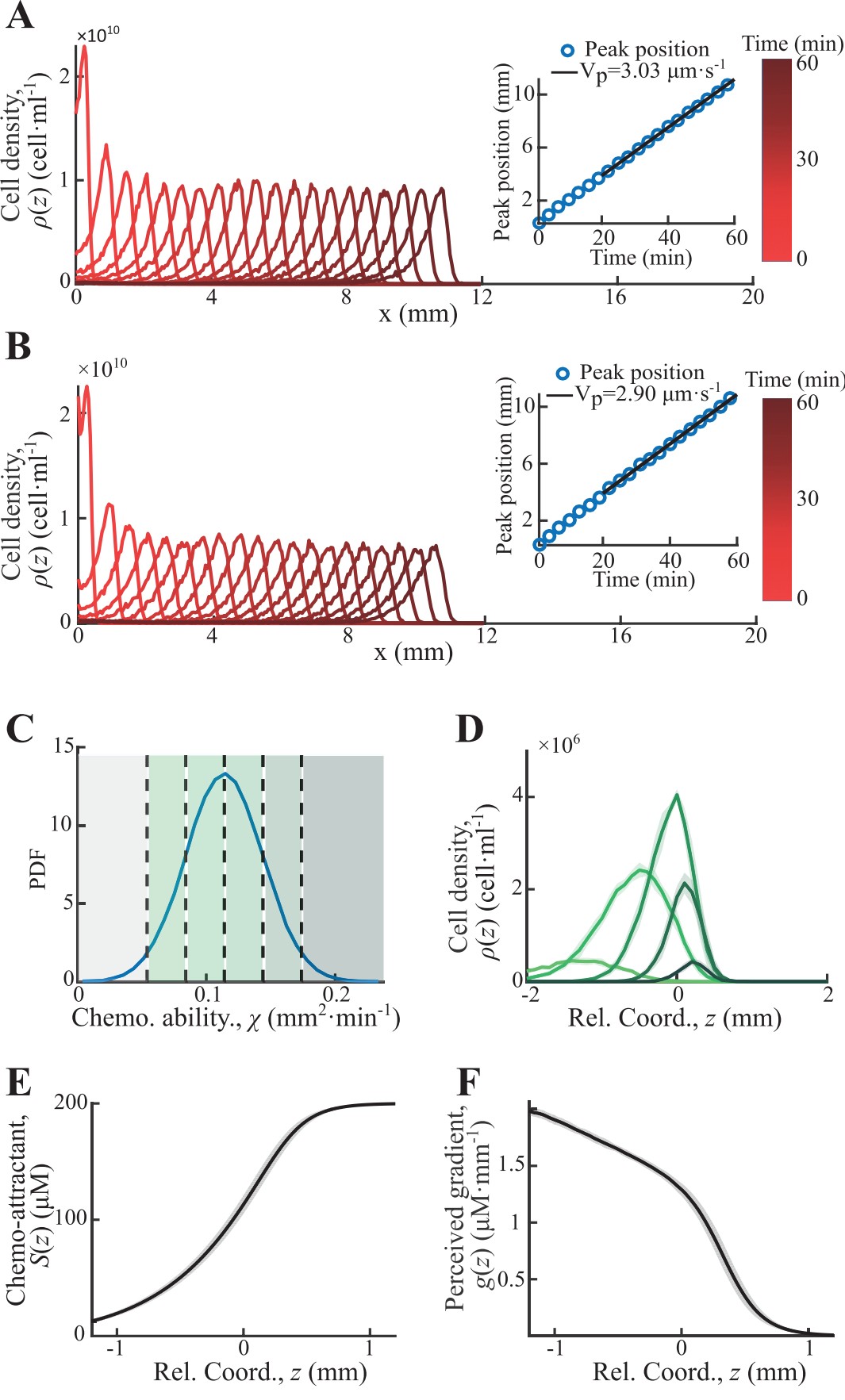

**Appendix 1—figure 5.** Particle-based simulation with chemoattractants consumption. (**A**) Simulated density profiles of active particles of single phenotype of $\chi = 0.11$ mm$^2$/min . $V_G = 3.0$ $\mu$m/s over time. The peak positions increase linearly with time (insert plot). (**B**) Simulated density profiles of active particles of heterogenous phenotypes distributed in (**C**). The peak positions follow a stable speed of $V_G = 2.9$ $\mu$m/s (insert plot). (**D**) Time-shifted density profiles of different phenotypes exhibit spatially ordered in moving coordinate. Cells in the gray area drop the group during migration. Time-shifted chemoattractant profile increases from back to front (**E**) while the perceived gradient decreases with a quasi-linear region around 0. (**F**) Shaded area in D, E, F represents s.d. of 40 profiles recorded at stable region (from 21 to 60 min) with time step of 1 min.

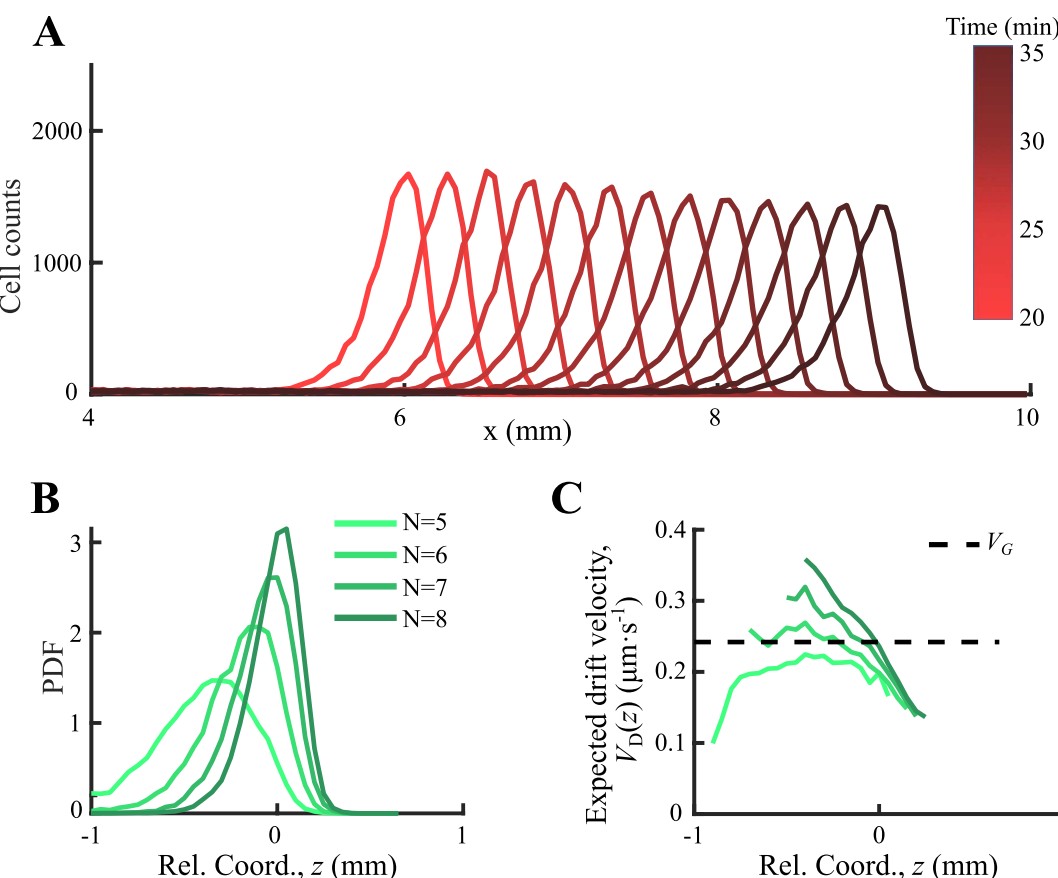

**Appendix 1—figure 6.** Agent-based simulations with consumption of chemoattractant. (**A**) Simulated density profiles of bacteria integrated with chemotactic pathway show stable migration after 20 min simulation. Tracks were obtained in from 30 to 32 min, and were calibrated into a moving coordinate through group migration speed. (**B**) Four subgroups of different receptor gain $N$ were orderly distributed. (**C**) The expected drift velocity $V_D(z)$ is decreasing and is spatially ordered.

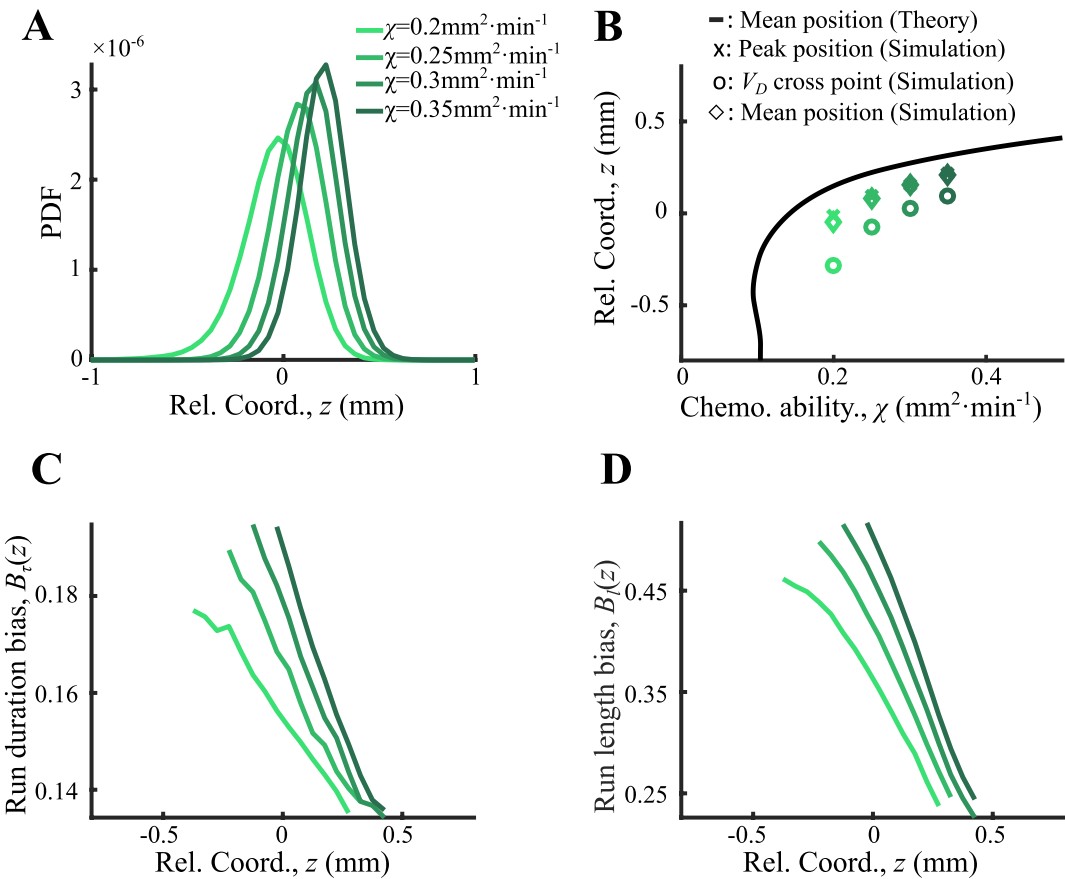

**Appendix 1—figure 7.** Agent-based simulations of cells with different receptor gains. (**A**) Bacteria of different chemotatic ability ($\chi$) form ordered structure in a moving chemoattractant field as in **Appendix 1—figure 4D** with fixed velocity $V_G = 3.0$ $\mu$m/s. (**B**) The average positions of each phenotype (diamonds), the peak positions (crosses), and the cross positions between their $V_D$ curves and the group velocity $V_G$ (circles) increase with $\chi$. These positions follow the same trend as predicted by Ornstein-Uhlenbeck (OU) model (black line). (**C, D**) Run duration bias $B_\tau$ and run length bias $B_l$ of different subgroups decrease from back to front. Both curves were also spatially ordered by $\chi$ (color from light to dark green).

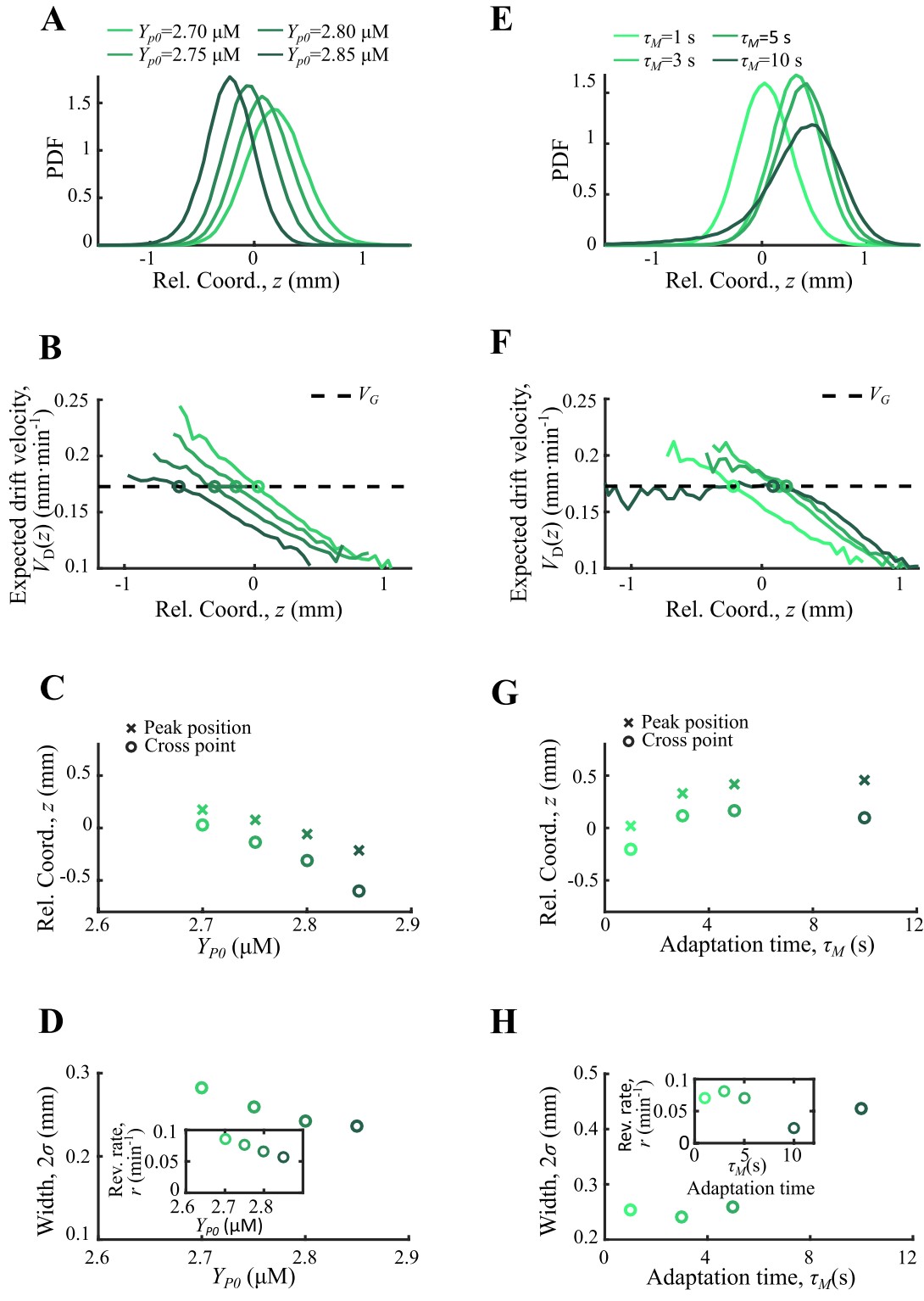

**Appendix 1—figure 8.** Agent-based simulations of cells with different CheY-P levels and adaptation times. Simulations results for CheY-P concentration $Y_{p0}$ ranging from 2.70 to 2.85 μM (A–D) and for adaptation time $\tau_M$ ranging from 1 to 10 s (E,H) are presented. In both cases the cell density $\rho(z)$ (A, E) and the expected drift velocity $V_D(z)$ (B, F) are spatially ordered according to the varying parameter. The mean position (crossed) together with the cross position between $V_D(z)$ and $V_G$ (circles) increase with $Y_{p0}$ (C) while they show non-monotonic trend to $\tau_M$ (G). The width of the density profile (defined

*Appendix 1—figure 8 continued on next page*

*Appendix 1—figure 8 continued*

by s.d. of all cells) and the reversion rate decrease with $Y_{p0}$ (D). And they showed a non-monotonic dependence on $\tau_M$ . The receptor gain $N$ is set to 24 in both cases, while $Y_{p0}$ is set to 2.77 μM for the $\tau_M$ varying case and $\tau_M$ is 1 s for the $Y_{p0}$ varying case. Simulations are performed under a steeper moving chemoattractant gradient of $g_1 = -2.5\,\mathrm{mm}^{-2}$ .

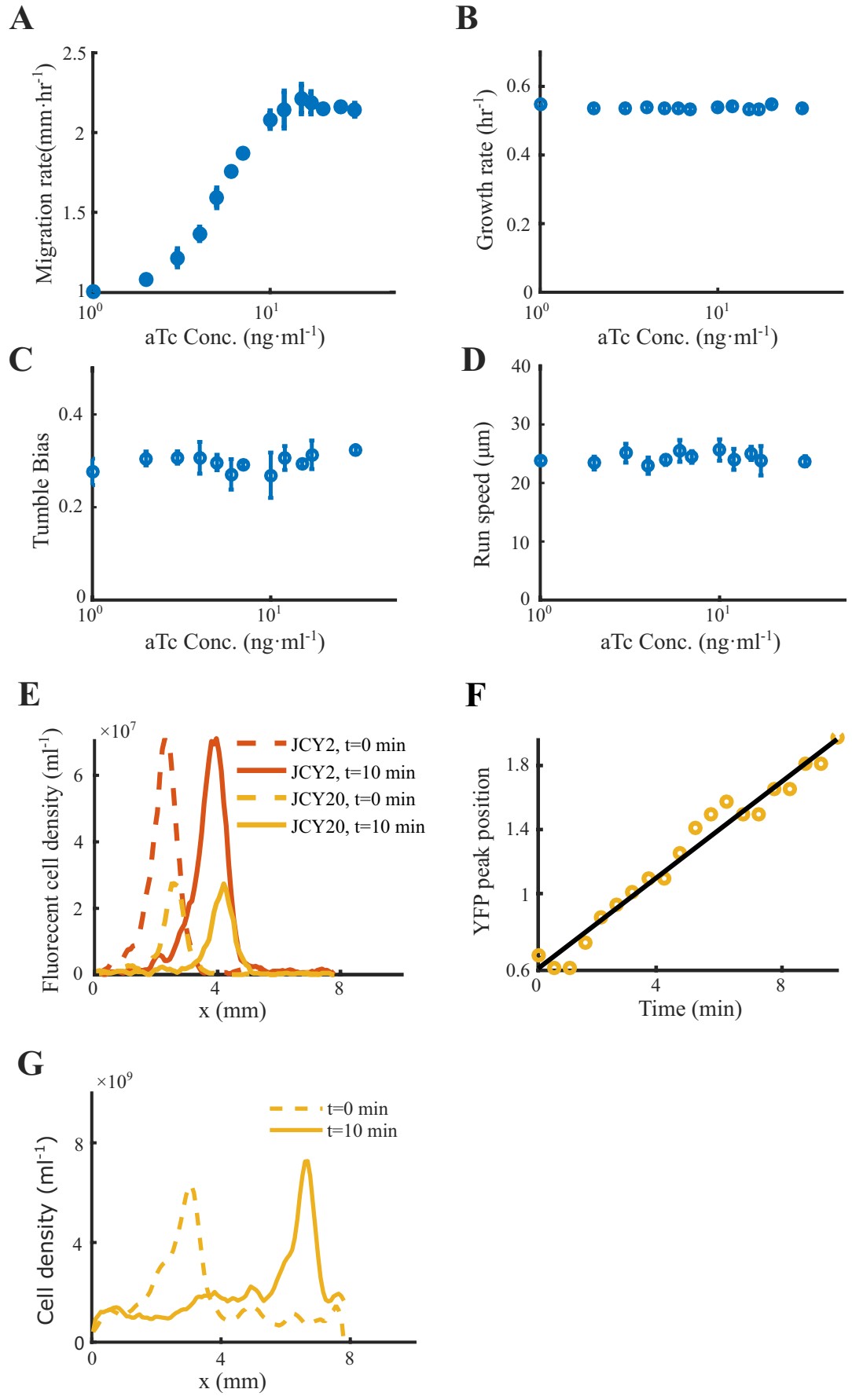

**Appendix 1—figure 9.** Migration rates, growth rates, motilies and coherent migration of the Tar-titratable strain. (**A**) Tar-titratable strain (JCY20) allows us to change its chemotaxis ability (measured by its migration rate on agar plate) by adding inducer ([aTc]) of different concentrations. (**B**) This strain has constant growth rates measured in liquid batch culture. (**C–D**) Motility statistics were extracted from bacterial tracks in free liquid. The growth rate, tumble bias, and mean run speed were found constant with all tested aTc concentrations. Error bar represents s.d. of three biological replicates. (**E**) The density profile of the wild-type (WT) strain with red fluorescent protein (RFP) (strain JCY2) (red lines) and Tar-titratable strain with yellow fluorescent protein (YFP) (strain JCY20) (yellow lines) are plotted over space at time before (dashed lines) and after tracking (solid lines). Because the density profiles of both strains before and after tracking are considered to be identical, the migration group is stable during the tracking period (10 min ). The fluorescent strains (JCY2 and JCY20) are mixed with WT non-fluorescent strain (RP437) by ratio of 8:1:400. (**F**) During the 10 min tracking period, only the Tar-titratable strain (JCY20) is recorded. For each half a minute, we plotted the density profiles and find their peak positions (yellow circles). Such peak positions increase linearly with time (fitted by the black line). Data were extracted from one experiment. (**G**) The density profile of the Tar-only strain with YFP (strain UU1624 from Prof. Parkinson) are plotted over space at time before (dashed lines) and after 10 min (solid lines).

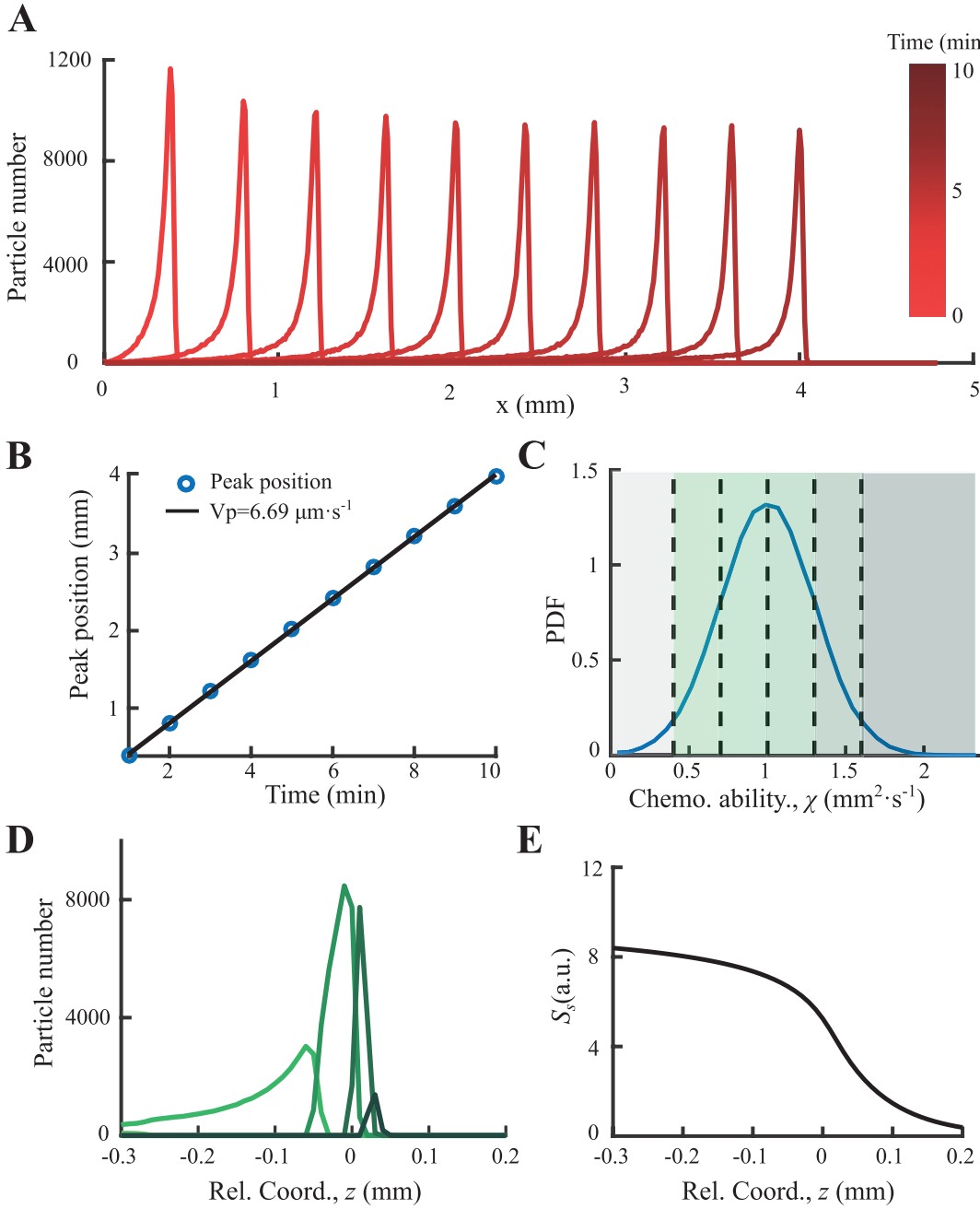

**Appendix 1—figure 10.** Particle based simulation of the signal secretion model. (**A**) Simulated density profiles of heterogenous phenotypes distributed in C show (**B**) stable group migration with its peak positions following a constant speed of $V_p$ = 6.69 $\mu$m/s. (**C**) The phenotypic diversity was set by a Gaussian distribution on $\chi$, with mean value of 1 and s.d. of 0.3. (**D**) Time-shifted density profiles of different phenotypes exhibit spatially ordered in moving coordinate. Color code follows C, while cells in the gray area drop from the group. (**E**) Time-shifted signal profile $S_s$, which plays the role of 'global driving force', decreases from back to front.

