## [Decision Letter]

**Acceptance summary:**

The authors present a study on the cohesion maintenance of *E. coli* during collective migration in a self-generated gradient. They performed experiments and complemented the study with a predictive model and simulation to understand how bacteria with different phenotype are able to move as a cohesive group and how the individual bacterium defines its own position within the group. Particularly interesting aspects of the study are the use of titration of behavior with chemoreceptor abundance and the use of potential wells to model the attraction of bacteria to the center of their cohesive group. This approach will be of interest to physicists and biologists interested in collective motility and migration.

**Decision letter after peer review:**

[Editors’ note: the authors submitted for reconsideration following the decision after peer review. What follows is the decision letter after the first round of review.]

Thank you for submitting your work entitled "Spatial modulation of individual behaviors enables collective decision-making during bacterial group migration" for consideration by *eLife*. Your article has been reviewed by 3 peer reviewers, and the evaluation has been overseen by a Reviewing Editor and a Senior Editor. The reviewers have opted to remain anonymous.

We are sorry to say that, after consultation with the reviewers, we have decided that your work will not be considered further for publication by *eLife*.

In this paper, Bai et al. investigate through experiments and agent-based modelling how cohesion is maintained in bacterial waves on chemotactic landscapes created by nutrient consumption. The manuscript confirms that the behavior of individuals is modulated in such a way that makes cells converge towards the center of the group. Behavioral modulation in different phenotypes ensures an ordered spatial arrangement of the phenotypes. All reviewers appreciate the careful experiments and data analysis as well as the introduction of a number of technical advancements (e.g. the titration of behavior with the chemo receptor abundance, and the formalization of bacterial attraction as a potential well). However, the main results appear to confirm previous findings (Saragosti et al. 2011, Fu et al. 2018). Thus, the level of insight provided by the analysis is not sufficient to grant publication in *eLife*.

*Reviewer #1:*

In this paper, Bai et al. investigate in experiments and simulations how cohesion is maintained in chemotactic travelling waves of bacteria. These waves emerge from the bacterial population consuming an attractant, thus carving a gradient which they follow chemotactically. This paper builds up on previous work of some of the authors (Fu et al., Nat Commun 2018), which found that in these waves bacteria with varying degree of chemotactic sensitivity organize spatially in the band, which allows for its cohesiveness despite varying phenotypes. The authors investigate here an additional element for the cohesiveness of the wave: because the sharpness of the gradient increases from the front to the back of the wave, 'late' cells catch up via a stronger chemotactic response, and front cells slow down via a weaker one. This had been already postulated in earlier work on the phenomenon (Saragosti et al. PNAS 2011), but here the authors investigate how this applies to cells with varying chemotactic sensitivity. They also performed agent-based simulations of the cells behavior in the gradient and developed a model of the motion in the gradient. The latter maps the spatial dependence of the gradient steepness onto an effective travelling potential which keeps the cells together in a group as the gradient and the wave propagate. Importantly, the effective potential is predicted to be tighter for cells with higher chemotactic sensitivity, in agreement with the cell behavior they observe in experiments where the chemotactic sensitivity is artificially modulated. This suggests that weakly chemotactic cells are more weakly bound to the group and have a higher chance of being left behind. This last part is interesting in the context of range extension in semi-solid agar, where bacteria are known to be spatially organized and selected according to their chemotactic motility (Ni et al., Cell reports 2017, Liu et al. Nature 2019).

This paper builds its strengths on the extensive experimental characterization of the system and a variety of modeling approaches and makes a fairly convincing case for the way of understanding the mechanism of cohesion maintenance they propose.

From a methodological perspective, only a few points need to be addressed:

Control experiments need to quantify the cell-to-cell variability of the induction level of Tar by tetracycline.

Chemical attraction to cues released by other cells is a well-documented way to create cohesive large scale structures in *E. coli* (Budrene and Berg Nature 1995, Park et al. PNAS 2003, Jani et al. Microbiology 2017, Laganenka et al. Nat commun 2016). The cohesion of the wave have never been analyzed in this optic, despite being a possible alternative explanation to the gradient shape. Since the authors main claim is about the wave cohesion, they should provide evidence that such an explanation can be ruled out or considered secondary.

Possible effects of physical interactions between cells on the chemotactic response are not accounted for. The consequences should be better discussed, because they are known to influence chemotactic motility at the densities encountered in the present experiments (Colin et al. Nat commun 2019).

Additionally, the paper could better emphasize the new results and separate them from the confirmations of previous results.

1. I would highly recommend a thorough correction of the English language. Although some parts are quite fine and require only minor fix, others can be very hard to read and understand. Even when the English is fine, streamlining the presentation of the results could also improve the read considerably.

2. The discussion and the abstract are the places to better separate between confirmation of previous results and new finding, to emphasize the new findings.

3. For the possible effect of chemical cues, simulations or experiments in a Tar-only strain could be good tests.

4. The maximal density in the peak seems to be about 1% volume fraction (10^10 cells/mL, Figure 1) in the experiments. At these densities, chemotaxis is known to be affected by physical interaction between cells (Colin et al. Nat commun 2019). I would suggest additional simulation were \chi is modulated according to local density following (Colin et al. Nat commun 2019) to test whether an effect is present.

5. I would suggest to explain why agent based simulations are necessary (memory effects, etc) after the particle based simulation.

6. L216 it would be a good idea to explain the conceptual difference between VD(z) and VI(z)(=VG), and why they differ, since this is central to the analysis, and might not be obvious to all readers.

*Reviewer #2:*

The manuscript by Bai et al. explores the single-cell motility dynamics within a chemotactic soliton wave in *E. coli*. They tracked individual cells and measured their trajectory speed and orientation distributions behind and ahead of the wave. They showed cells behind the wave were moving in a more directed fashion towards the center of the wave compared to cells ahead of the wave. This behavior explains the stability of group migration, as confirmed by numerical simulations.

I do not recommend this manuscript for publication in *eLife* since it basically reproduces and deepens previous published works. In particular, Saragosti et al. (2011) already provided exactly what the authors claim to do here : "How individuals with phenotypic and behavioral variations manage to maintain the consistent group performance and determine their relative positions in the group is still a mystery." (Line 75-77) (See the last sentences from Saragosti et al. : "This modulation of the reorientations significantly improves the efficiency of the collective migration. Moreover, these two quantities are spatially modulated along the concentration profile. We recover quantitatively these microscopic and macroscopic observations with a dedicated kinetic model.")

What is novel here is the titration of the behavior with chemo-receptor abundance, but I believe the scope is not wide enough for publication in *eLife*. I suggest the authors to submit in a more specialized journal.

The authors should make more explicit what is really new in their work, compared to what is already known. In the present form, it is hard to pinpoint exactly the novelty of this research.

*Reviewer #3:*

The authors present a study on the collective behaviour of *E. coli* during migration in a self-generated gradient. Taking into account phenotypic variation within a biological population, they performed experiments and complemented the study with a predictive model used for simulation to understand how bacteria can move as a group and how the individual bacterium defines its own position within the group.

They observed experimentally that phenotype variation within the bacterial population causes a spatial distribution within the chemotactic band that is not continuous but formed by subpopulations with specific properties such as run length, run duration, angular distribution of trajectories, drift velocity. They attribute this behaviour to the chemotaxis ability, which varies between phenotypes and defines a potential well that anchors each bacterium in its own group. This was proven by the subdiffusive dynamics of the bacteria in each subgroup. Many cases were studied in the experiments and the authors present many controls to clearly demonstrate their hypothesis.

These are interesting results that prove how a discretised distribution can produce continuous collective behaviour. It presents also an interesting example in the field of active matter about collective behaviour on a large scale that is generated by a different behaviour of individuals on a much smaller scale. However, it is not clear how the subpopulations can be held together in the group. Moreover, a link between bacterial dynamics and the biological necessary mechanism is not clear.

They formulate a theoretical description based on the classical Keller-Segel model. Langevin dynamics was used to describe bacterial activity in terms of drift velocity for simulation, which agrees very well with experimental observations.

One can appreciate the interesting results of the study describing Ecoli chemotaxis as a mean-reversion process with an associated potential, but it is not clear to what extent the results can be generalised to all bacteria or rather relate to the strain the authors investigated.

1) In the Results section, lines 93-181, the authors show the results of their experiments, which essentially confirm the results of previous studies in terms of the average speed of the group and the distribution of running length and running duration from back to front within the group, as well as the angular distribution of running length. I have difficulty seeing the differences between this work and the previous studies. In fact, other studies already showed the persistence of the cell migration pathway from the back to the front as well as cells migrating faster in the back and slower in the front.

The manuscript would benefit greatly from a clear comparison between the authors' results and the previous studies.

2) However, they noted that the tumble bias is constant and not spatially modulated. This is the first difference compared to the previous studies cited, and it would be useful to have a guess or speculation about the physiological significance of this. Is the tumble bias related to the bacterial strain? Shouldn't the tumble bias be a strategy of the organism to scan the environment? Do the authors believe that tumble bias is intrinsic to the system and cannot be influenced by physiological priorities such as receptor occupancy and foraging? In the previous study, tumble bias caused faster migration in the posterior region and slower migration in the anterior region. The authors observed that in their case, the faster migration in the back and slower migration in the front was due to drift speed. How do they explain this difference in these observations?

Such aspects should be clarified if the authors intend to claim that their outcomes advance the knowledge in the field of bacterial migration otherwise they are rather considering a subcase, a special Ecoli bacteria strain.

3) The authors propose a discretisation of the chemotactic band into subgroups whose dimension is defined by the chemotactic ability with an inverse relationship to the SD γ of the bacterial distribution. Based on this idea, they suggest that each group represents a potential well. Although the idea is very interesting, it is not entirely clear to me how the bacteria can reverse to the mean of the group just because they rely on the molecular migration pathway. How is the attraction to each group generated? Would it make sense to think about the mechanism of quorum sensing, which Ecoli bacteria are known to use for population sensing? This would also explain how the exit of each group is avoided: the chemotactic ability pulls the bacteria towards the gradient, but the quorum sensing, e.g. a population sensing, drives the mechanism towards the group. This means that the driving force that causes the group to move together is the sum of at least two contributions.

4) Linked to the previous question: How are the different subpopulations kept together? Is a difference in drift velocity within a range between the back and the front sufficient to prevent the entire chemotactic group from disintegrated? Have the authors tested other drift velocity ranges to see if there is a threshold for these group dynamics? What about accounting for a molecular response?

5) In line 360, the authors claim to obtain the same subdiffusive behaviour of the bacteria when the migratory ability is influenced by the adaptation time or the basal CheY protein level. From the supplementary material, one can understand that this was the result of the simulation. For this reason, the authors should be very careful when claiming that they have observed how different proteins influence the modulation of behaviour. It is not clear from the simulation how this result can be clearly attributed to the specific protein CheY. One can choose a different protein and simulate the behaviour and get the same result without having any connection to the biological real state. I suggest that the authors explain more about this point or remove it from the text and just leave it in the supplementary material with a clear explanation about the missing connection with the biology and clarify that this is a speculation.

6) In line 104, the authors explain that the band forms after centrifugation. Their simulation shows that this happens after 20 min. What about the experiments? Is there consistency between simulation and experiments?

[Editors’ note: further revisions were suggested prior to acceptance, as described below.]

Thank you for submitting your article "Spatial modulation of individual behaviors enables an ordered structure of diverse phenotypes during bacterial group migration" for consideration by *eLife*. Your article has been reviewed by 3 peer reviewers, and the evaluation has been overseen by a Reviewing Editor and Aleksandra Walczak as the Senior Editor. The reviewers have opted to remain anonymous.

The revised manuscript addresses more clearly the novelty of the work. However, some concerns remain over novelty and new concepts introduced within the revision.

Essential revisions:

1) One main novelty the authors claim with respect to Fu et al. is they propose a mechanism for the ordering. Please clarify whether this is different from the mechanism proposed by Fu et al.

2) Explain that the wave travels because of attractant consumption. Verify with numerical simulations that the ordering persists even when the gradient is continuously modified by consumption.

3) Clarify what a pushed wave precisely is.

Please address all other points raised by the individual referees as you see fit.

*Reviewer #1:*

In their appeal, the authors have rewritten the text to make it significantly clearer and put the work better in context with previous publications. They also addressed my technical concerns. The main reason for rejection was however the lack of sufficient novelty. The two main points of the paper are:

1) Bringing evidence of a mechanism of wave coherence at fixed chemotactic sensitivity by an increased drift of the late cells and a reduced drift for the early ones thanks to the shape of the gradient, which the authors called mean-reversion or now pushed wave-front. This mechanism was already heavily suggested by the results of Saragosti et al. 2011 and proposed as a mechanism in that paper. On this point, I acknowledge that the presentation and analysis of the cell behavior in this paper does a better and more thorough job at demonstrating the phenomenon than the previous one, and the authors do show that this coherence holds for different values of the chemotactic sensitivity. It however remains that the results simply confirm the previously inferred mechanism, using the same experimental technique.

2) Explaining how spatial ordering allows the reconciliation of phenotypic variability and a coherent wave-front. On this point, I do not think the authors bring any new information compared to Fu et al. (2018). For instance, the mechanism for spatial ordering of the mean position of the various spatial phenotypes is already very well illustrated by Figure 3a of that paper.

The authors also reemphasized the importance of the gradient shape in maintaining the coherence of the wave. Here, and contrary to Fu 2018, they however systematically took the gradient shape as a given during simulations and did not investigate its emergence from consumption by the heterogeneous population. This diminishes the interest of the paper by this much.

All in all, I maintain my appreciation that this is technically a work of quality but its findings still remain fairly incremental, and I think it could be best suited for a more specialist journal.

*Reviewer #2:*

In the revised version of the manuscript, the authors satisfactorily added significant pieces of data to the whole story. They explained why their work differs from previously published data (Saragosti at all, 2011).

They improved the logical flow of the text (presentation of tracking data, stochastic modeling, agent-based modeling, titration), which now better pinpoints what is novel. They added a stochastic model to better understand the mechanisms underlying group migration.

Therefore, I recommend this manuscript for publication in *eLife*, provided that the authors can answer the following point.

Could the authors explain what a pushed wave is? Pushed wave/pulled wave have a clear meaning in the context of traveling waves (FKPP reaction and variants). Briefly, a pulled wave is when the per capita growth rate is the highest at the edge of the front. A pushed wave is when the per capita growth rate is the highest behind the front. Here, cells move but do not divide. This should be clarified.

*Reviewer #3:*

In this new version of the manuscript, authors Bai et al. offer a rewording of the text that greatly improves the understanding of their study.

They provide a new abstract that helps to explain the innovation of their results and their relevance to the biological event of cell migration.

They expand the text by adding details about their experiments, how they confirm the theoretical model and how these can improve our knowledge on collective motion of bacteria. They explain why their results are able to answer the open questions left by the studies of Saragosti 2011 and Fu 2018. In this way they discuss the differences between their study and the previous ones.

Overall, the text and the improved figures allow one to appreciate the originality of her study. Specially on the following points:

1) The analysis at the level of the individual cell and how individual behaviour can lead to collective migratory behaviour;

2) The importance of decreasing drift velocity within the chemotactic band for the collective migration of bacteria as a group;

3) The coexistence of phenotypic variability within the same migratory population and the formulation of the potential well hypothesis to explain group cohesion;

5) The adequacy of the titrated phenotype control experiment, which may also suggest a possible molecular pathway involved in the process.

The new version is able to convey the importance of the study, which is robust from an experimental point of view, with several control experiments that leave no doubt about the hypothesis that the authors draw from their observations and that are used to confirm their theoretical model. All the concerns I expressed were satisfactorily addressed.

I would suggest improving the text further so that some repetitions and mistakes are removed to make it more easy to read, and then I would suggest the manuscript for publication.

---

## [Author Response]

[Editors’ note: The authors appealed the original decision. What follows is the authors’ response to the first round of review.]

Reviewer #1:In this paper, Bai et al. investigate in experiments and simulations how cohesion is maintained in chemotactic travelling waves of bacteria. These waves emerge from the bacterial population consuming an attractant, thus carving a gradient which they follow chemotactically. This paper builds up on previous work of some of the authors (Fu et al., Nat Commun 2018), which found that in these waves bacteria with varying degree of chemotactic sensitivity organize spatially in the band, which allows for its cohesiveness despite varying phenotypes. The authors investigate here an additional element for the cohesiveness of the wave: because the sharpness of the gradient increases from the front to the back of the wave, 'late' cells catch up via a stronger chemotactic response, and front cells slow down via a weaker one. This had been already postulated in earlier work on the phenomenon (Saragosti et al. PNAS 2011), but here the authors investigate how this applies to cells with varying chemotactic sensitivity. They also performed agent-based simulations of the cells behavior in the gradient and developed a model of the motion in the gradient. The latter maps the spatial dependence of the gradient steepness onto an effective travelling potential which keeps the cells together in a group as the gradient and the wave propagate. Importantly, the effective potential is predicted to be tighter for cells with higher chemotactic sensitivity, in agreement with the cell behavior they observe in experiments where the chemotactic sensitivity is artificially modulated. This suggests that weakly chemotactic cells are more weakly bound to the group and have a higher chance of being left behind. This last part is interesting in the context of range extension in semi-solid agar, where bacteria are known to be spatially organized and selected according to their chemotactic motility (Ni et al., Cell reports 2017, Liu et al. Nature 2019)This paper builds its strengths on the extensive experimental characterization of the system and a variety of modeling approaches and makes a fairly convincing case for the way of understanding the mechanism of cohesion maintenance they propose.

In fact, we have addressed both the mechanism to maintain a coherent group and also the mechanism to form ordered pattern of diverse phenotypes. Thanks to the reviewer, we noticed that the second point was not clearly showed out in our previous version. So that we have largely rewritten the texts and reorganized the results to prominent both mechanism.

From a methodological perspective, only a few points need to be addressed:Control experiments need to quantify the cell-to-cell variability of the induction level of Tar by tetracycline.

The distributions of the titrate cells are presented by a ptet-Tar-GFP strain, where the GFP is used as a reporter of the expressed Tar protein. The results are shown in Author response image 1.

**Author response image 1. sa2fig1:** 

Chemical attraction to cues released by other cells is a well-documented way to create cohesive large scale structures in *E. coli* (Budrene and Berg Nature 1995, Park et al. PNAS 2003, Jani et al. Microbiology 2017, Laganenka et al. Nat commun 2016). The cohesion of the wave has never been analyzed in this optic, despite being a possible alternative explanation to the gradient shape. Since the authors main claim is about the wave cohesion, they should provide evidence that such an explanation can be ruled out or considered secondary.

We thank the reviewer to point out the self-attractant secretion as a possible mechanism to maintain coherent group. We argue that this mechanism is not necessary for the chemotactic group to maintain coherency, because the migration group keeps without considering these effects in our agent based simulations.

Moreover, as suggested by the reviewer, we Used a Tar only strain, which do not sense any chemo-attractant other than aspartate, to show that the migration group maintained coherent (see Figure S9). This experiment showed that the secretion of self-attractant is not essential for the coherent group migration.

Possible effects of physical interactions between cells on the chemotactic response are not accounted for. The consequences should be better discussed, because they are known to influence chemotactic motility at the densities encountered in the present experiments (Colin et al. Nat commun 2019).

As being reported by Colin et al., the effective drift velocity and the chemotactic ability deceases when cells are condensed (volume fraction >0.01). However, the cell density is smaller than this critical value (volume fraction<0.01).

Additionally, the paper could better emphasize the new results and separate them from the confirmations of previous results.

In the revised version, we addressed 2 new findings:

1. The individual drift velocity decreases from back to front of the bacterial migration group, which makes the chemotactic migration wave a pushed wave.

Cells of diversed phenotypes follows the same reversion behavior, ie. drift faster in the back and slower in the front, but with ordered mean positions, to achieve the ordered pattern in the migration group.

1. I would highly recommend a thorough correction of the English language. Although some parts are quite fine and require only minor fix, others can be very hard to read and understand. Even when the English is fine, streamlining the presentation of the results could also improve the read considerably.

The texts are largely rewritten and the English language is edited by a professional English editor.

2. The discussion and the abstract are the places to better separate between confirmation of previous results and new finding, to emphasize the new findings.

Abstract and discussion are rewritten to emphasize the new findings:

1. The drift velocities of individual bacteria decrease from the back to the front.2. The individual run-and-tumble random motions are spatially modulated, and enables the bacterial population to migrate as a pushed wave.3. The pushed wave can help a diverse population to stay in a tight group.4. Diverse individuals perform the same type of mean reverting processes around centers orderly aligned by their chemotactic abilities.

3. For the possible effect of chemical cues, simulations or experiments in a Tar-only strain could be good tests.

Our agent-based simulation was based on an single chemical cue and uses Tar receptor only. New experiments is performed with a Tar-only strain provide by Prof. Johan Sandy Parkinson. Coherent group migration was observed, related results were added in the Figure S9.

4. The maximal density in the peak seems to be about 1% volume fraction (10^10 cells/mL, Figure 1) in the experiments. At these densities, chemotaxis is known to be affected by physical interaction between cells (Colin et al. Nat commun 2019). I would suggest additional simulation were \chi is modulated according to local density following (Colin et al. Nat commun 2019) to test whether an effect is present.

The effect of interaction between cells can be neglected. According to Colin et al. Nat commun 2019, the chemotactic drift velocity and the chemotaxis coefficient of bacterial are almost constant for volume fraction <0.01. Since the volume fraction in our case maximize in 0.01, the cell density in the migrating band can be considered as a constant.

5. I would suggest to explain why agent based simulations are necessary (memory effects, etc) after the particle based simulation.

Addressed in the main text, see Line 326-330.

6. L216 it would be a good idea to explain the conceptual difference between VD(z) and VI(z)(=VG), and why they differ, since this is central to the analysis, and might not be obvious to all readers.

Addressed in the main text, see Line 148-156.

Reviewer #2:The manuscript by Bai et al. explores the single-cell motility dynamics within a chemotactic soliton wave in *E. coli*. They tracked individual cells and measured their trajectory speed and orientation distributions behind and ahead of the wave. They showed cells behind the wave were moving in a more directed fashion towards the center of the wave compared to cells ahead of the wave. This behavior explains the stability of group migration, as confirmed by numerical simulations.I do not recommend this manuscript for publication in eLife since it basically reproduces and deepens previous published works. In particular, Saragosti et al. (2011) already provided exactly what the authors claim to do here : "How individuals with phenotypic and behavioral variations manage to maintain the consistent group performance and determine their relative positions in the group is still a mystery." (Line 75-77) (See the last sentences from Saragosti et al. : "This modulation of the reorientations significantly improves the efficiency of the collective migration. Moreover, these two quantities are spatially modulated along the concentration profile. We recover quantitatively these microscopic and macroscopic observations with a dedicated kinetic model.")

Saragosti et al. talks about the modulation of reorientation angle of bacteria along directions. It is not equal to the spatial modulation of drift velocities along space. They claim that cells moving along the gradient direction reorient less during a tumble than cells moving against the gradient. This phenomenon increases the migration efficiency of the group. Here, in our paper, we claim that the drift velocity of bacteria is spatially modulated, where cells on the back drifts faster while the cells in the front drift slower. This phenomenon is important because it makes the chemotactic migration front a pushed wave, that helps the group to keep diversed phenotypes.

Although Saragosti et al. Have also suggested spatial modulation of bias in run length to explain the coherency of the migration group. But they did not quantify such bias nor did they explain the causes and consequences of the spatial modulation. Moreover, Their model consisting their proposed mechanism of directional persistence, cannot explain their observed phenomenon of the decreasing bias of run length (see figure 4A and C).In this circumstance, we can’t agree that they already proofed how cells with diversed phenotype to maintain coherent group.

Moreover, they did not talk about diversities in the group.

Saragosti et al. Figure 4 shows that the theoretical predictions (green and red lines in C) do not match the experiment measurement (green and red lines in A).

What is novel here is the titration of the behavior with chemo-receptor abundance, but I believe the scope is not wide enough for publication in eLife. I suggest the authors to submit in a more specialized journal.

The titration of the chemo-receptor abundance of bacteria serves as a tool to explain how diverse individuals manage to form the ordered patterns in a group. This question worth several discussion because diversity is known as an important feature to keep a group to survive. The ordered pattern was found the key for a migrating group to keep the diversity while performing consistent migration speed. In this paper we successfully explained how individuals performing biased random walk are able to form ordered structure.

The authors should make more explicit what is really new in their work, compared to what is already known. In the present form, it is hard to pinpoint exactly the novelty of this research.

We notice that our previous layout of the results didn’t highlight the novelty of this paper. So that we have reorganized the results and have largely revised the texts to address the new findings.

Reviewer #3:The authors present a study on the collective behaviour of *E. coli* during migration in a self-generated gradient. Taking into account phenotypic variation within a biological population, they performed experiments and complemented the study with a predictive model used for simulation to understand how bacteria can move as a group and how the individual bacterium defines its own position within the group.They observed experimentally that phenotype variation within the bacterial population causes a spatial distribution within the chemotactic band that is not continuous but formed by subpopulations with specific properties such as run length, run duration, angular distribution of trajectories, drift velocity. They attribute this behaviour to the chemotaxis ability, which varies between phenotypes and defines a potential well that anchors each bacterium in its own group. This was proven by the subdiffusive dynamics of the bacteria in each subgroup. Many cases were studied in the experiments and the authors present many controls to clearly demonstrate their hypothesis.These are interesting results that prove how a discretised distribution can produce continuous collective behaviour. It presents also an interesting example in the field of active matter about collective behaviour on a large scale that is generated by a different behaviour of individuals on a much smaller scale. However, it is not clear how the subpopulations can be held together in the group.

The decreasing chemo-attractant gradient makes the migration wavefront a pushed wavefront. So that the balanced position of the subpopulation with larger chemotactic ability is located in the front where the gradient is small. So that diverse phenotypes form ordered pattern to achieve identical migration speed on their balanced positions. This discussion was added in the revised text (see line 268-277).

Moreover, a link between bacterial dynamics and the biological necessary mechanism is not clear.

The bacterial individual dynamics is controlled by the bacterial chemotaxis pathway, which is clear according to previous studies. Basically, the biased random motion was controlled by alternating expected run length through a temporal comparison mechanism between received chemo-attractant concentrations.(Jiang et al. 2010 Plos Comp. Biol.)

They formulate a theoretical description based on the classical Keller-Segel model. Langevin dynamics was used to describe bacterial activity in terms of drift velocity for simulation, which agrees very well with experimental observations.One can appreciate the interesting results of the study describing Ecoli chemotaxis as a mean-reversion process with an associated potential, but it is not clear to what extent the results can be generalised to all bacteria or rather relate to the strain the authors investigated.

The mean reversion process is a result of decreasing drift velocity (or a pushed wave). Although our study focuses on bacterail chemotaxis migration, but the ordering mechanism of diversed phenotypes follows a OU type model, which is not limited to bacterial chemotaxis. In this case, we argue that the ordering mechanism that we proposed is universal to all active particles that generate signals as a global cue of collective motion.

1) In the Results section, lines 93-181, the authors show the results of their experiments, which essentially confirm the results of previous studies in terms of the average speed of the group and the distribution of running length and running duration from back to front within the group, as well as the angular distribution of running length. I have difficulty seeing the differences between this work and the previous studies. In fact, other studies already showed the persistence of the cell migration pathway from the back to the front as well as cells migrating faster in the back and slower in the front.The manuscript would benefit greatly from a clear comparison between the authors' results and the previous studies.

Thank for the valuable suggestion. We realized that our previous layout of the results buries our key idea and mislead the reader to confuse our results with previous studies (especially the study of Saragosti et al.). So that we have largely reorganized our results to highlight the ordering mechanism of diversed phenotypes, that none of the previous studies have proposed before.

In fact, although Saragosti et al. have proposed the scenario of reversion behavior in bacterial chemotaxis group migration, they did not prove it through experimental quantification nor numerical simulations. The have proposed the dynamics of directional persistence that bacterial reorient less during a tumble event if they are migrating to the gradient direction. However, this dynamic is insufficient to explain the spatial distribution of run lengths nor the bias of it

2) However, they noted that the tumble bias is constant and not spatially modulated. This is the first difference compared to the previous studies cited, and it would be useful to have a guess or speculation about the physiological significance of this. Is the tumble bias related to the bacterial strain? Shouldn't the tumble bias be a strategy of the organism to scan the environment? Do the authors believe that tumble bias is intrinsic to the system and cannot be influenced by physiological priorities such as receptor occupancy and foraging? In the previous study, tumble bias caused faster migration in the posterior region and slower migration in the anterior region. The authors observed that in their case, the faster migration in the back and slower migration in the front was due to drift speed. How do they explain this difference in these observations?

We didn’t observe different tumble bias profile compared to previous studies. It has a decreasing profile from back to front, indicating that cells of better motility are located in the front. This phenomenon is also consistent with our agent-based simulations (Figure S8).

Such aspects should be clarified if the authors intend to claim that their outcomes advance the knowledge in the field of bacterial migration otherwise they are rather considering a subcase, a special Ecoli bacteria strain.

The tumble bais is not constant over space, since the run time increase form back to front while the tumble time keeps constant. Our experiment supports the fact that tumble bias decrease from back to front. This phenomenon is a result of both phenotypic ordering and the spatial modulation, and the spatial profile of tumble bias depends on the distribution of phenotypes in the group. The tumble bias is intrinsic however, it is also affected by the environment through the adaptation process. (Fu et al. 2018).

3) The authors propose a discretisation of the chemotactic band into subgroups whose dimension is defined by the chemotactic ability with an inverse relationship to the SD γ of the bacterial distribution. Based on this idea, they suggest that each group represents a potential well. Although the idea is very interesting, it is not entirely clear to me how the bacteria can reverse to the mean of the group just because they rely on the molecular migration pathway. How is the attraction to each group generated?

The potential well is generated by two opposite forces, the decreasing gradient of chemo-attractant that pushes the cells to catch up the group and relative motion of the group itself that lefts cells to fall behind.

Would it make sense to think about the mechanism of quorum sensing, which Ecoli bacteria are known to use for population sensing? This would also explain how the exit of each group is avoided: the chemotactic ability pulls the bacteria towards the gradient, but the quorum sensing, e.g. a population sensing, drives the mechanism towards the group. This means that the driving force that causes the group to move together is the sum of at least two contributions.

As you may know, previous studies have shown that bacterial may secret self-attractant or auto-inducer to modify their motions. However, the collective migration doesn’t rely on this effect because theories (Keller et al. 1973) and simulations may easily repeat the collective migration without considering these effects.

Moreover, we used a Tar only strain which senses only aspatate as chemo-attractant, may form the same coherent collective migration as the wild type strain. New results were added in Figure S9, and it further confirms the effect of self-attractant and quorum sensing can be neglected. This discussion was added in the main text, see line 473-476.

4) Linked to the previous question: How are the different subpopulations kept together? Is a difference in drift velocity within a range between the back and the front sufficient to prevent the entire chemotactic group from disintegrated? Have the authors tested other drift velocity ranges to see if there is a threshold for these group dynamics? What about accounting for a molecular response?

The decreasing profiles of the drift velocity makes sure that the driving force of the migration wave has a decreasing form (also known as a pushed wave), In a pushed wave, subpopulations are able to keep moving coherently. To Proof this, we simulated sub populations migration from pushed wave to pulled waves, corresponding to the diving force from decreasing form to increasing form. The results confirmed that pushed wave maintains diversity, these results were added to figure 2.

However, in the real system, the decreasing drift velocity has an upper bound on the back of the wave, so that the height of the potential well that pushes bacteria to move forward is limited. So that cells are continuously left behind. See discussions in line 500-511

5) In line 360, the authors claim to obtain the same subdiffusive behaviour of the bacteria when the migratory ability is influenced by the adaptation time or the basal CheY protein level. From the supplementary material, one can understand that this was the result of the simulation. For this reason, the authors should be very careful when claiming that they have observed how different proteins influence the modulation of behaviour.

Good suggestion, we have clarified it in the new version (see line 363)

It is not clear from the simulation how this result can be clearly attributed to the specific protein CheY. One can choose a different protein and simulate the behaviour and get the same result without having any connection to the biological real state. I suggest that the authors explain more about this point or remove it from the text and just leave it in the supplementary material with a clear explanation about the missing connection with the biology and clarify that this is a speculation.

The agent based simulation is constructed based on knowledge of bacterial chemotaxis signaling pathway. Thanks to the previous studies (Jiang, Qi et al. 2010, Sneddon, Pontius et al. 2012), the molecular mechanism is quantitatively established. The agent based model we use in this text is modified from previous models that has been well demonstrated (Dufour, Fu et al. 2014, Long 2019). So that we have the confidence to say that the parameters in the simulation is related to specific proteins of clear biological meaning.

6) In line 104, the authors explain that the band forms after centrifugation. Their simulation shows that this happens after 20 min. What about the experiments? Is there consistency between simulation and experiments?

The experiments were performed in half an hour after sample loading, which is comparable to our simulations. This information is added in the protocol part. (see line 599).

[Editors’ note: what follows is the authors’ response to the second round of review.]

Essential revisions:1) One main novelty the authors claim with respect to Fu et al. is they propose a mechanism for the ordering. Please clarify whether this is different from the mechanism proposed by Fu et al.

We thank for this advice. To clarify the novelty of this study, we revised the discussion in Line459-470, and specifically discussed our new findings in respect to Fu et al.

Fu et al. first discovered the ordered structure of cells with different phenotypes during group migration. By classic Keller-Segel model, Fu et al. has proposed a phenomenological explanation that the maintenance of group speed over different phenotypes is self-consistent with the constant product of chemotactic ability and perceived gradient. The KS model predicts that perceived gradient is high in the back and low in the front, which suggests the cells maintain an opposite order in terms of chemotactic abilities. However, Fu et al. did not provide a direct measurement of gradient shape, nor explain how individual bacteria (as small as micrometer scales) behave so as to determine their relative positions in the migration group (in millimeter scales).

In this study, our study bridges from single-cell behaviors to population coherent performance, and separate the contributions of phenotypic diversity and dynamic behavioral modulation to group structure, further clarifying the explanation proposed by Fu et al. By analyzing single cell trajectories during group migration, we discovered the spatially structured profile of drift velocity. This finding is a direct evidence how individual behaviors are modulated when imposed to the perceived gradient during the group migration. The decreasing profile of drift velocity plays a role of driving force, which enables the migratory population to propagate as a pushed wavefront. We further demonstrated that the pushed wave can maintain cells with diversity in a compact group, and individuals behave as mean reversion relative to the group. Moreover, cells with phenotypic diversity exhibit the same shape of drift velocities, but perform mean reverting processes around centers spatially aligned based on their chemotactic abilities. Therefore, we demonstrated that population order can emerge from diverse individuals following the same rule of behavioral modulation.

2) Explain that the wave travels because of attractant consumption. Verify with numerical simulations that the ordering persists even when the gradient is continuously modified by consumption.

Thanks for the advice. We added an explanation that the attractant consumption was the cause of the traveling wave in the revised introduction text line 72.

The agent-based simulation with attractant consumption was already presented in FigS6, showing that the ordering remains. However, this was not emphasized in the previous version. In the revised version, we addressed both the numerical methods to solve the attractant consumption equation and relating results in the revised text line 334-345.

Moreover, the numerical simulation based on the Langevin-type model with attractant consumption may also confirm that the ordering effect is independent to the source of the decreasing moving gradient (see Figure S5 and main text line 263-265).

3) Clarify what a pushed wave precisely is.

A pushed wave or pulled wave is originally defined by the driving force profile across a front of a traveling wave. As their names have inferred a ‘pushed’ wave has its maximal driving force in the back while a ‘pulled’ wave has its maximal driving force in the front [van Saarloos, 2003]. In the context of the diffusion-growth system, the traveling wave known as the Fisher wave or F-KPP wave, the criterion of pushed or pulled wave is translated into the profile of per capita growth rate because the driving force of such traveling wave is the population growth, so that a decreasing or increasing profile of the growth rate is used as the criterion of pushed or pulled wave. The definition of the pushed wave is given in the revised text line 237-240. And a discussion of the pulled wave in the F-KPP case is added in the revised text line 489-493.

Reviewer #1:In their appeal, the authors have rewritten the text to make it significantly clearer and put the work better in context with previous publications. They also addressed my technical concerns. The main reason for rejection was however the lack of sufficient novelty. The two main points of the paper are:1) Bringing evidence of a mechanism of wave coherence at fixed chemotactic sensitivity by an increased drift of the late cells and a reduced drift for the early ones thanks to the shape of the gradient, which the authors called mean-reversion or now pushed wave-front. This mechanism was already heavily suggested by the results of Saragosti et al. 2011 and proposed as a mechanism in that paper. On this point, I acknowledge that the presentation and analysis of the cell behavior in this paper does a better and more thorough job at demonstrating the phenomenon than the previous one, and the authors do show that this coherence holds for different values of the chemotactic sensitivity. It however remains that the results simply confirm the previously inferred mechanism, using the same experimental technique.

We thank the reviewer for the appreciation of the quality of our work. In this study, we improved our single cell tracking technique, enabling us to obtain long trajectories. This improvement helps us to get the spatial profile of the drift velocity which is not reported previously. Based on this observation, we further established a pushed wavefront picture of population propagation emerging from behavioral modulation of individual cells. This further allowed us to explain the mechanism of the ordering phenomenon from the single cell perspective. Although Saragosti et al. 2011 has suggested a similar picture (the mean-reversion), we don’t think their proposed model (e.g. the directional persistence) has explained the reversion behavior.

2) Explaining how spatial ordering allows the reconciliation of phenotypic variability and a coherent wave-front. On this point, I do not think the authors bring any new information compared to Fu et al. (2018). For instance, the mechanism for spatial ordering of the mean position of the various spatial phenotypes is already very well illustrated by Figure 3a of that paper.

Fu et al. first discovered the ordered structure of cells with different phenotypes during group migration. They have proposed a phenomenological explanation that the maintenance of group speed over different phenotypes is self-consistent with the constant product of chemotactic ability and perceived gradient (Figure 3a). However, this phenomenological explanation did not provide a direct measurement of gradient shape, nor how individual bacteria (as small as micrometer scales) behave so as to determine their relative positions in the migration group (in millimeter scales).

In this study, we provided a clear mechanism how this population order arises from individual random motions. The measured single cell behavior and the drift velocity profile is a direct evidence how individual behavior are modulated when imposed to the perceived gradient during the group migration. And the decreasing profile of drift velocity plays a role of driving force, which enables the migratory population to propagate as a pushed wavefront. We further demonstrated that the pushed wave can maintain cells with diversity in a compact group, and individuals behave as mean reversion relative to the group. Moreover, cells with phenotypic diversity exhibit the same shape of drift velocities, but perform mean reverting processes around centers spatially aligned based on their chemotactic abilities. Therefore, our study further clarifies the explanation proposed by *Fu et al.*, and proposed a novel mechanism of population order emerging from diverse individuals with stochastic behaviors.

The authors also reemphasized the importance of the gradient shape in maintaining the coherence of the wave. Here, and contrary to Fu 2018, they however systematically took the gradient shape as a given during simulations and did not investigate its emergence from consumption by the heterogeneous population. This diminishes the interest of the paper by this much.

We have taken the measured gradient shape to simplify the analysis of the OU-type model, and to emphasize that our proposed ordering mechanism are extendable to all the pushed waves. But we did investigate the emergence of the gradient from population consumption by both the Langevin type model (Figure S5), and the simulation of the agent-based model (Figure S6). Moreover, we have extended the emergence of a decreasing global force from population consumption to population secretion (Figure S10) to generalize our proposed ordering mechanism.

Reviewer #2:In the revised version of the manuscript, the authors satisfactorily added significant pieces of data to the whole story. They explained why their work differs from previously published data (Saragosti at all, 2011).They improved the logical flow of the text (presentation of tracking data, stochastic modeling, agent-based modeling, titration), which now better pinpoints what is novel. They added a stochastic model to better understand the mechanisms underlying group migration.Therefore, I recommend this manuscript for publication in eLife, provided that the authors can answer the following point.

We thank the reviewer for the recommendation.

Could the authors explain what a pushed wave is? Pushed wave/pulled wave have a clear meaning in the context of traveling waves (FKPP reaction and variants). Briefly, a pulled wave is when the per capita growth rate is the highest at the edge of the front. A pushed wave is when the per capita growth rate is the highest behind the front. Here, cells move but do not divide. This should be clarified.

Yes, the cells in our case does divide, but we can still define a pushed wave in its original definition: a ‘pushed’ wave has its maximal driving force in the back while a ‘pulled’ wave has its maximal driving force in the front of a traveling wave [van Saarloos, 2003]. In the context of the diffusion-growth system, known as the Fisher wave or F-KPP wave, the original criterion of pushed or pulled wave was translated into the position of the per capita growth rate. This is because the driving force of such traveling wave is the population growth, so that a decreasing or increasing profile of the growth rate is used as the criterion of pushed or pulled wave.

Reviewer #3:In this new version of the manuscript, authors Bai et al. offer a rewording of the text that greatly improves the understanding of their study.They provide a new abstract that helps to explain the innovation of their results and their relevance to the biological event of cell migration.They expand the text by adding details about their experiments, how they confirm the theoretical model and how these can improve our knowledge on collective motion of bacteria. They explain why their results are able to answer the open questions left by the studies of Saragosti 2011 and Fu 2018. In this way they discuss the differences between their study and the previous ones.Overall, the text and the improved figures allow one to appreciate the originality of her study. Specially on the following points:1) The analysis at the level of the individual cell and how individual behaviour can lead to collective migratory behaviour;2) The importance of decreasing drift velocity within the chemotactic band for the collective migration of bacteria as a group;3) The coexistence of phenotypic variability within the same migratory population and the formulation of the potential well hypothesis to explain group cohesion;5) The adequacy of the titrated phenotype control experiment, which may also suggest a possible molecular pathway involved in the process.The new version is able to convey the importance of the study, which is robust from an experimental point of view, with several control experiments that leave no doubt about the hypothesis that the authors draw from their observations and that are used to confirm their theoretical model. All the concerns I expressed were satisfactorily addressed.I would suggest improving the text further so that some repetitions and mistakes are removed to make it more easy to read, and then I would suggest the manuscript for publication.

Thanks for the appreciation and the advice. In the revised version, the typos are corrected and phrases are revised. Main figures are reformatted to be clear.